# Incorporation of multi-phase halogen chemistry into Community Multiscale Air Quality (CMAQ) model

**Kiyeon Kim[1], Chul Han Song[1*], Kyung Man Han[1], Greg Yarwood[2], Ross Beardsley[2], and Saewung Kim[3]**

1. School of Earth Science and Environmental Engineering, Gwangju Institute of Science and Technology (GIST), Gwangju 61005, Republic of Korea
2. Ramboll, Novato, CA 94945, USA
3. Department of Earth System Science, University of California, Irvine, CA, USA

**Shortened title:** Multi-phase halogen chemistry

**\*Corresponding author:** Chul Han Song (chsong@gist.ac.kr)

(Submitted to *Atmospheric Chemistry and Physics*)

**Abstract**

Halogen radicals (Cl, Br, and I) significantly influence atmospheric oxidation capacity, affecting both $O_3$ formation and destruction. However, understanding of halogen chemistry remains limited. To better investigate comprehensive atmospheric halogen chemistry, we incorporated halogen processes into the Community Multi-scale Air Quality (CMAQ) model: (i) emissions of $Cl_2$, HCl, $Br_2$, and HBr from anthropogenic sources, and $Br_2$, $I_2$, HOI, and halocarbons from natural sources; and (ii) 177 multi-phase halogen reactions. Model performance was evaluated against observed $ClNO_2$ levels, and by comparison with reported ranges of BrO and IO levels. The updated model showed significant improvements in simulating $ClNO_2$ mixing ratios, with the index of agreement (IOA) increasing from 0.41 to 0.66 and mean bias (MB) decreasing from $–159.36$ ppt to -25.07 ppt at supersites. Furthermore, simulated BrO and IO levels fell within the ranges reported in previous studies. We found that these improvements were driven by four key reactions: (i) ClO self-reaction, (ii) heterogeneous HOBr chemistry, (iii) $NO_2$ uptake, and (iv) revised $N_2O_5$ parameterization. Based on our modeling system, we found that the presence of halogen radicals led to changes in the net $O_x$ production rate ($P(O_x)$), which increased from 3.08 ppb/h to 3.33 ppb/h on land and decreased from 0.21 ppb/h to 0.07 ppb/h over ocean. It was noted that levels of OH, HCHO, and $NO_x$ also increased by ~0.007 ppt (5.5%), ~0.03 ppb (1.6%), and ~0.29 ppb (2.9%), respectively, while levels of $HO_2$ and VOCs decreased by ~0.45 ppt (5.3%) and ~0.71 ppb (5.9%). These results highlight the importance of accurately representing halogen processes in regional air quality models.

**Keywords:** Halogen chemistry; Anthropogenic halogen emission; Nitryl chloride ($ClNO_2$); Net $O_x$ production ($P(O_x)$); Regional air quality;

## 1. Introduction

Atmospheric oxidants, such as OH, $NO_3$, and $O_3$, play a significant role in atmospheric chemistry. These oxidants react with volatile organic compounds (VOCs), leading to the formation of peroxyl radicals ($RO_2$), which, in turn, influences the $O_3$ formation. They also contribute to the formation of secondary organic and inorganic aerosols. Meanwhile, halogen radicals (such as Cl, Br, and I) also serve as oxidants in the atmosphere, affecting the oxidation capacity through various reactions (R1 – R4; see below) (Simpson et al., 2015; von Glasow and Crutzen, 2003; Fan and Li, 2022).

$$X \text{ (Cl, Br, and I)} + VOC \ \rightarrow \ RO_2 \tag{R1}$$

$$RO_2 + NO \ \rightarrow \ NO_2 + RO \tag{R2}$$

$$HO_2 + NO \ \rightarrow \ NO_2 + OH \tag{R3}$$

$$NO_2 \overset{hv}{\rightarrow} O^{3p} + O_2 \overset{M}{\rightarrow} O_3 \tag{R4}$$

$$X + O_3 \ \rightarrow \ XO + O_2 \tag{R5}$$

$$XO + HO_2 \ \rightarrow \ HOX + O_2 \tag{R6}$$

$$HOX \overset{hv}{\rightarrow} X + OH \tag{R7}$$

These radicals can also make substantial impacts on the $O_3$ loss via reactions (R5) – (R7) (Saiz-Lopez et al., 2012; Sarwar et al., 2015; Simpson et al., 2015). Given their roles in both the $O_3$ formation and destruction, a comprehensive understanding of atmospheric halogen chemistry is essential for accurately assessing the oxidative potentials of the atmosphere.

In this context, several studies attempted to incorporate chlorine chemistry into chemical-transport models (e.g., Yi et al., 2021; Sarwar et al., 2012; Qiu et al., 2019a and b). Specifically, Qiu et al. (2019a) reported that heterogeneous reactions involving reactive chlorine species can increase $O_3$ levels by approximately 20%. Moreover, Liu et al. (2018) found that the mixing ratios of $O_3$ increased by ~7.7 ppbv when anthropogenic chlorine emissions were included.

On the other hand, numerous studies have emphasized not only the significance of chlorine chemistry but also the influence of the synergistic effects of bromine and iodine chemistry with chlorine chemistry (Sarwar et al., 2019; Simpson et al., 2015; Caram et al., 2023; Badia et al., 2019; Saiz-Lopez et al., 2014; Li et al., 2022; Iglesias-Suarez et al., 2020). For instance, modeling studies considering both bromine and iodine chemistries showed that simulated $O_3$ levels actually decreased by 15.9 ppb (Parrella et al., 2012; Sarwar et al., 2015; Herrmann et al., 2022; Gantt et al., 2017; Read et al., 2008; Huang et al., 2021). These findings strongly suggest that incorporating chlorine processes, together with bromine and iodine processes, is crucial for correct and comprehensive understanding of atmospheric chemistry.

The Korean Peninsula, surrounded by the Yellow Sea, Korea Strait, and the East Sea, is characterized by high population density and highly industrial regions. Therefore, it can be influenced by both natural (oceanic) and anthropogenic halogen emissions. However, almost no modeling study has taken into account the natural and anthropogenic halogen processes over/around South Korea. Although almost no research has been carried out to examine the impacts of halogen chemistry on atmospheric composition over/around South Korea, several studies have considered atmospheric chlorine processes using 3D chemical-transport models (CTMs) (e.g., Jo et al., 2023; Kim et al., 2023). Given the synergistic effects of chlorine, bromine and iodine chemistries in the atmosphere, a comprehensive study that takes all these halogen processes into account is absolutely necessary.

For the comprehensive analysis of halogen processes and their influence on regional air quality, we established anthropogenic and natural halogen emissions and incorporated full sets of halogen reactions into the framework of the Community Multi-scale Air Quality (CMAQ) model. Specifically, we incorporated the following halogen processes into the CMAQ model: (i) atmospheric chlorine processes (anthropogenic HCl and $Cl_2$ emissions with 58 chlorine reactions); (ii) atmospheric bromine processes (anthropogenic and natural HBr and

Br$_2$ emissions together with 64 bromine reactions); and (iii) atmospheric iodine processes (HOI
and I$_2$ natural emissions, along with 55 iodine reactions).

Based on this modeling system, the primary objectives of this study are threefold: (i)

to develop and implement an updated halogen chemistry that accounts for the interactions
among multiple halogen species; (ii) to evaluate model performance using observational data
from the Korea US Air Quality (KORUS-AQ) campaign (1 May – 12 June, 2016), including
direct ClNO$_2$ measurements and inferred estimates of BrO and IO; and (iii) to examine the
atmospheric impacts of the updated halogen chemistry on a broader set of key atmospheric
constituents such as O$_3$, OH, HO$_2$, HCHO, VOC$_s$, and NO$_x$. In this context, this study is
designed to support a comprehensive understanding of atmospheric halogen processes and their
implications for regional air quality.

**2. Methodology**

In this study, we incorporated homogeneous, aqueous, and heterogeneous halogen

reactions into the CMAQ model, along with emissions of halogen species. To evaluate the
accuracy of these halogen processes, we compared the model results with observational data
from the KORUS-AQ campaign (Jeong et al., 2019; Crawford et al., 2021). This section
provides several details on the observation data, the WRF-CMAQ model configurations, and
the atmospheric halogen processes, including halogen reactions and emissions.
**2.1 Observation data**

Mixing ratios of nitryl chloride (ClNO$_2$) were measured every five minutes at Olympic

Park (37.52°N; 127.12°E) and Mt. Taewha (37.27°N; 127.41°E) stations during the period of
the KORUS-AQ campaign (refer to two blue stars in Fig. 1a), using Chemical Ionization Mass
Spectrometer (CIMS). The CIMS instrument has a detection limit of 1.5 ppt and an uncertainty
within 20%. Further details on the CIMS instrument are found in Slusher et al. (2004) and
Jeong et al. (2019). In our study, $ClNO_2$ observations were utilized to evaluate the performance
of the modified CMAQ model simulations. These results are discussed in Sect. 3.1.

**2.2 WRF-CMAQ model description**

The Weather Research and Forecasting (WRF) v3.8.1 model simulations were carried
out to generate meteorological fields (Skamarock et al., 2008). The details of physical
parameters used in the WRF simulations are summarized in Table S1. National Center for
Environmental Prediction Final Analysis (NCEP-FNL) data were used for initial and boundary
conditions. The WRF model included a 5-day spin-up period to minimize uncertainties from
the initial and boundary conditions.
This study also included the CMAQ v5.2.1 model simulations (Byun and Schere, 2006)
over a domain covering northeast Asia with $273 \times 204$ horizontal grid cells. The grid
resolution is $15 \times 15$ km$^2$ with 15 vertical layers from surface to 50 hPa. The Statewide Air
Pollution Research Center-07 (SAPRC-07TC) mechanism (Carter, 2010; Hutzell et al., 2012)
with AERO6 module was used in the CMAQ model simulations. One limitation of the SAPRC-
07TC mechanism is that it has only basic chlorine chemistry. In order to implement more
sophisticated halogen model simulations, we incorporated additional and updated halogen
reactions into the SAPRC-07TC mechanism. The detailed reactions are explained in Sects.
2.4.1 and 2.4.2.
In order to run the CMAQ model, biomass burning and biogenic emissions were
obtained from the Fire Inventory from NCAR (FINN) v1.5 (Wiedinmyer et al., 2011) and the
Model of Emissions of Gases and Aerosol from Nature (MEGAN) v2.1 (Guenther et al., 2012),
respectively. Anthropogenic emissions were acquired from the KORUS v5.0 emission
inventory (Woo et al., 2020), specifically developed for the KORUS-AQ campaign. The
KORUS v5.0 inventory covers emissions of primary pollutants such as $NO_x$, CO, HCHO,
VOCs, and particulate chlorine (pCl$^-$), but it omits the emissions for anthropogenic chlorine
species (HCl and $Cl_2$), bromine species (HBr and $Br_2$), and ocean-generated halogen species
(HOI, $I_2$, and halocarbons). We have thus developed new halogen emissions for this study. The
methodology for developing the halogen emissions will be discussed in Sections 2.3.1 and

2.3.2.

**2.3 Halogen emissions**
In this section, we discuss the development of anthropogenic and natural halogen
emissions within our model framework.
**2.3.1 Anthropogenic emissions**
First, we assumed that emissions of anthropogenic HCl and $Cl_2$ mainly originated from
coal combustion. Coal combustion occurs predominantly in four main sectors: industry,
residential areas, power plants, and other sectors such as agriculture and furniture
manufacturing. In addition, HCl emissions also take place from municipal solid waste
incineration.
To calculate chlorine emissions from industry and residential areas, we utilized coal
consumption data from the 2016 Regional Energy Report of South Korea
(https://www.keei.re.kr). Thereafter, the emissions of HCl and $Cl_2$ from these sectors were
calculated using the following equation (1):

$$E_{i,j} = M_{i,j} \times EF_{i,j} \times \rho \times \frac{1}{MM} \times \frac{1}{10^3} \qquad \text{(Eq. 1)}$$

where $E_{i,j}$ represents the emission for species $i$ in categories $j$ (Mg); $M$ denotes the coal
consumption (Gg); and $EF$ is the emission factor (μg/g) calculated using the method from a
previous study (Jiang et al., 2005 (in Chinese)). $\rho$ indicates the percentage of HCl and $Cl_2$ in
the chlorine content of coal. In our study, percentages of 86.3% and 3.63% for $\rho_{HCl}$ and $\rho_{Cl2}$
were used, respectively, based on research conducted by Deng et al. (2014) and Liu et al. (2018).
$MM$ represents the ratios of the molar mass of the chlorine atom to the molecular weight (i.e.,
35.5/36.5 for HCl and 1 for $Cl_2$).

For the remaining three sectors, namely power plants, solid waste incineration, and

others, the HCl emissions were obtained directly from the Korean tele-monitoring system
(TMS) named the CleanSYS (https://cleansys.or.kr). Meanwhile, the emissions of $Cl_2$ from
these sectors can also be calculated using equation (1), based on the HCl emissions previously
calculated.

Bromine emissions (HBr and $Br_2$) were additionally estimated from the previously

calculated chlorine emissions. According to a recent study, bromine is also emitted from the
coal combustion with a ratio (0.25) of bromine to chlorine concentrations (Peng and Wu, 2014).
These bromine emissions were split into 70% and 30% for HBr and $Br_2$, respectively. The
detailed methodology used in our study is summarized in Li et al. (2021).

Consequently, we developed an emission inventory that includes anthropogenic

chlorine and bromine emissions. Figure 1 illustrates the spatial distributions of these emissions
across South Korea. The total emission rates for anthropogenic HCl, $Cl_2$, HBr, and $Br_2$ in South
Korea are 5,989.6, 450.8, 460.8, and 240.8 $Mg \cdot yr^{-1}$ respectively. These values are higher than
those reported in previous studies conducted for the same region and time period. For instance,
Jo et al. (2023) estimated that total annual anthropogenic HCl emissions were less than 1.0 Gg,
whereas Kim et al. (2023) reported a value of 1.35 Gg. These discrepancies may result from
the inclusion of additional HCl emissions from the residential and industrial sectors in our study.
Moreover, our research also accounts for emissions of $Cl_2$, HBr, and $Br_2$. It is noteworthy that
$Cl_2$, HBr, and $Br_2$ have relatively shorter e-folding lifetimes (a few minutes for $Cl_2$ and $Br_2$,
and a few hours for HBr) than HCl (about 1.5 days), which may increase the oxidant capacity
in the atmosphere.

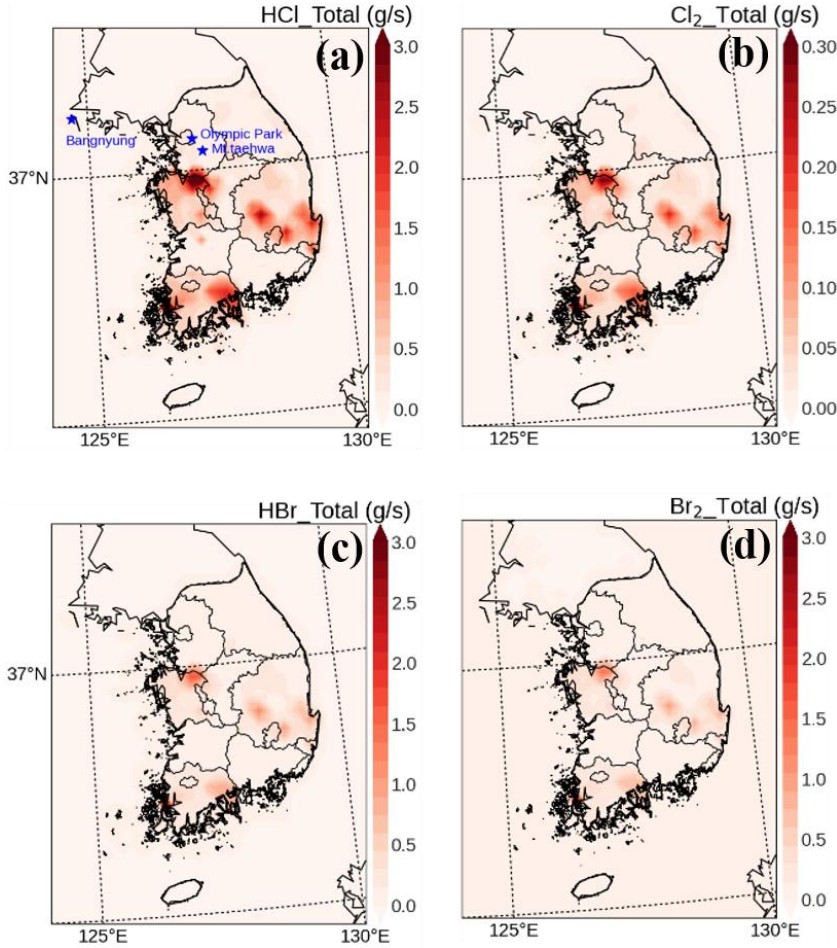


**Figure 1.** Spatial distributions of anthropogenic emission rates of (a) HCl, (b) $Cl_2$, (c) HBr, and (d) $Br_2$ during the period of the KORUS-AQ campaign over South Korea. Three blue stars denote the locations of three supersites (Bangnyung, Olympic Park, and Mt. Taehwa) during the KORUS-AQ campaign.

**2.3.2 Natural emissions**

Biogenic halocarbons, such as $CHBr_3$, $CH_2Br_2$, $CHBrCl_2$, $CH_2BrCl$, $CHBr_2Cl$, $CH_2I_2$, $CH_3I$, $CH_2ICl$, and $CH_2IBr$, are emitted from micro-algae activities in the ocean. To calculate these emissions of bromine and iodine-containing halocarbon species, we used the chlorophyll-a (chl-a) concentrations as a proxy for the photosynthetic activity of phytoplankton (Liss et al., 2014).

Chlorophyll-a concentrations in the ocean have been monitored by various satellite sensors such as Moderate Resolution Imaging Spectroradiometer (MODIS) and Geostationary Ocean Color Imager (GOCI) (Kim et al., 2016; O'reilly and Werdell, 2019; Sarwar et al., 2015).

Among these sensors, Park et al. (2015) reported that the chlorophyll-a concentrations
measured by the GOCI sensor showed the best agreement with surface observations compared
to those from the MODIS sensor in the East Asian ocean. Based on this study, we applied the
chlorophyll-a data from the GOCI sensor into our study (refer to Fig. S1).

Gridded halocarbon emissions were estimated using the following equation (2)

(Sarwar et al., 2015):

$E_{Halocarbon} = 1.2 \times 10^{-11} \times (O_F + S_F) \times A_{GC} \times f_{HC} \times f_{DP} \times [chl\text{-}a]$        (Eq.2)

where $O_F$ and $S_F$ represent the ocean and coastal fractions of the grid cell, respectively. $A_{GC}$
denotes the area of the grid cell (m$^2$). $f_{HC}$ is the emission factor for the species; and $f_{DP}$
represents the diurnal profile. [$chl\text{-}a$] denotes the chlorophyll-a concentration. The distributions
of natural bromine and iodine emissions are shown in Fig. S2.

Natural $Br_2$ emissions were estimated by debromination of sea-salt aerosols (SSAs),

following the method proposed by Sarwar et al. (2015) and Yang et al. (2005). In this approach,
$Br^-$ contained within the SSAs was released into the atmosphere as $Br_2$ (i.e., dehalogenation of
SSAs). This process was parameterized based on sea-salt aerosol mass, Br/NaCl mass ratio,
sea surface temperature, and 10-m wind speed. In addition, inorganic iodine (such as HOI and
$I_2$) emissions were calculated at the air-sea water interfaces, utilizing information on dry
deposition of $O_3$ over the ocean. This approach is based on the fact that the formations of HOI
and $I_2$ are initiated by reaction between iodide ($I^-$(aq)) and $O_3$ at the ocean surfaces. More
detailed information on natural inorganic emissions can be found in Sarwar et al. (2015).
**2.4 Halogen chemical reactions**

As mentioned previously, the conventional CMAQ v5.2.1 model accounts for only a

limited set of chlorine-related processes: (i) reactions between VOCs and Cl radicals (Sander
et al., 2010), (ii) a simplified reactive chlorine cycling mechanism (Sander et al., 2010), and
(iii) the uptake of $N_2O_5$ on the chloride-containing particles (Bertram and Thornton, 2009).
Therefore, we attempted to incorporate the multi-phase halogen reactions into the CMAQ
v5.2.1 model to investigate the influences of atmospheric halogen chemistry. Detailed
descriptions of these reactions are provided in the subsequent sections.
**2.4.1 Chlorine reactions**
The chlorine-related reactions were incorporated into the framework of SAPRC07-TC
mechanism. The chlorine reactions consist of: (i) adjusted reaction rate coefficients for 14
reactions (refer to R9 to R22 in Table 1); (ii) updated 29 gaseous chlorine reactions (refer to
R23 to R51 in Table 1); (iii) added two aqueous-phase reactions (refer to R1 to R2 in Table 2);
and (iv) incorporated four heterogeneous reactions involving three reactive halogen species,
HOCl, ClNO$_2$, and ClONO$_2$, with NO$_2$ partitioning onto chloride-containing particles (refer to
R2 to R6 in Table 3).
The parameterization of $\gamma_{N2O5}$ currently embedded in the CMAQ v5.2.1 model
(shown in R8) has not been greatly satisfactory for reproducing the atmospheric levels of
ClNO$_2$.
$$N_2O_5 + (1-\varphi)H_2O + \varphi Cl^{-1} \xrightarrow{\gamma_{N2O5}} (2-\varphi)HNO_3 + \varphi ClNO_2 \qquad (R8)$$
where $\varphi$ represents the yield of ClNO$_2$ as a function of the concentration of particulate
chloride [Cl$^-$] and aerosol water content [H$_2$O]. The calculation of $\varphi$ was proposed by Bertram
and Thornton (2009):
$$\varphi = \frac{1}{1+\frac{[H2O]}{483[Cl-]}} \qquad (Eq.\ 3)$$
Although the use of R8 and Eq. 3 have been an advance in considering the production
of ClNO$_2$ from chlorine-containing particles (Bertram and Thornton, 2009), several studies
have reported that the parameterizations of $\gamma_{N2O5}$ with R8 and Eq. 3 tend to produce excessive
amounts of nitrate and ClNO$_2$ (Riedel et al., 2012a; Li et al., 2016; Yu et al., 2020). This
overestimation may result from uncertainties in calculating aerosol water content ([H$_2$O])
estimated from the aerosol thermodynamic module (Chang et al., 2016). To deal with this issue,
several studies have explored different parameterizations of $\gamma_{N2O5}$. However, the $\gamma_{N2O5}$ still
appears to be over-estimated in many cases (e.g., Chang et al., 2016; Liu et al., 2019; Mcduffie
et al., 2018; Riedel et al., 2012a; Wang et al., 2017). In this context, we alternatively selected
a different parameterization for $\gamma_{N2O5}$ (see equations (4) to (7)). These parameterizations
were suggested by Riemer et al. (2003) and Evans and Jacob (2005):
$$\gamma_{N2O5} = f \times \gamma_1 + (1- f) \times \gamma_2 \qquad \text{(Eq. 4)}$$

$$f = \frac{m_{sulfate}}{m_{sulfate} + m_{nitrate}} \qquad \text{(Eq. 5)}$$

$$\gamma_1 = \alpha \times 10^{\beta} \qquad \text{(Eq. 6)}$$

$$\gamma_2 = 0.1 \times \gamma_1 \qquad \text{(Eq. 7)}$$

where $\alpha$ is set to be $2.79 \times 10^{-4} + 1.3 \times 10^{-4} \times RH - 3.43 \times 10^{-4} \times RH^2 + 7.52 \times 10^{-8} \times RH^3$.
$\beta$ is set at 0.48, when T < 282 K, or at $4 \times 10^{-2} \times (294 - T)$, when $T \geq 282$ K. $m_i$ represents
the aerosol mass concentration of species i. Complete lists of chlorine reactions embedded into
the SAPRC07TC mechanism are shown in Tables 1, 2, and 3.
**Table 1.** List of homogeneous chlorine reactions used in this study.

| No | Reaction | Reaction rate | Reference |
|---|---|---|---|
| R1 | $Cl_2 \xrightarrow{hv} 2Cl$ | #1.0/<Cl$_2$> | 1 |
| R2 | $ClNO \xrightarrow{hv} Cl + NO$ | #1.0/<ClNO> | 1 |
| R3 | $Cl + NO_2 + M = ClONO$ | 1.3e-30^-2.0&1.0e-10^-1.0&-1.0&0.6&1.0 | 1 |
| R4 | $Cl + NO_2 + M = ClNO_2$ | 1.8e-31^-2.0&1.0e-10^-1.0&-1.0&0.6&1.0 | 1 |
| R5 | $ClONO \xrightarrow{hv} Cl + NO_2$ | #1.0/<ClONO> | 1 |
| R6 | $Cl + NO_3 = ClO + NO_2$ | $2.4 \times 10^{-11}$ | 1 |
| R7 | $ClO + NO_2 = ClONO_2$ | 1.80e-31^-3.4&1.5e-11^-1.90&0.6&1.0 | 1 |
| R8 | $HOCl \xrightarrow{hv} Cl + OH$ | #1.0/<HOCl> | 1 |
| R9 | $ClONO_2 = Cl + NO_3$ | #1.0/<ClONO$_{2\_1}$> | 2 |
| R10 | $ClONO_2 = ClO + NO_2$ | #1.0/<ClONO$_{2\_2}$> | 2 |
| R11 | $Cl + NO + M = 2ClNO$ | $7.70 \times 10^{-32}(\frac{T}{300})^{-1.8}$ | 2 |
| R12 | $ClNO_2 \xrightarrow{hv} Cl + NO_2$ | #1.0/<ClNO$_2$> | 2 |
| R13 | $Cl + HO_2 = HCl$ | $1.4 \times 10^{-11}e^{270/T}$ | 2 |
| R14 | $Cl + HO_2 = ClO + OH$ | $3.6 \times 10^{-11}e^{-375/T}$ | 2 |
| R15 | $Cl + O_3 = ClO$ | $2.3 \times 10^{-11}e^{200/T}$ | 2 |
| R16 | $ClO + NO = Cl + NO_2$ | $6.4 \times 10^{-12}e^{290/T}$ | 2 |
| R17 | $ClONO_2 = ClO + NO_2$ | $2.0 \times 10^{-21}$ | 2 |
| R18 | $Cl + ClONO_2 = Cl_2 + NO_3$ | $6.5 \times 10^{-12}e^{135/T}$ | 2 |
| R19 | $ClO + HO_2 = HOCl$ | $2.6 \times 10^{-12}e^{290/T}$ | 2 |
| R20 | $ClO + ClO = Cl_2 + O_2$ | $1.0 \times 10^{-11}e^{-1590/T}$ | 2 |
| R21 | $OH + HCl = Cl$ | $1.8 \times 10^{-12}e^{-250/T}$ | 2 |
| R22 | $Cl + H_2 = HCl + HO_2$ | $3.1 \times 10^{-11}e^{-2270/T}$ | 2 |
| R23 | $ClO + O = Cl + O_2$ | $2.8 \times 10^{-11}e^{85/T}$ | 2 |
| R24 | $ClO + OH = Cl + HO_2$ | $7.4 \times 10^{-12}e^{270/T}$ | 2 |
| R25 | $ClO + OH = HCl + O_2$ | $6.0 \times 10^{-13}e^{230/T}$ | 2 |
| R26 | $HOCl + OH = ClO + H_2O$ | $3.0 \times 10^{-12}e^{-500/T}$ | 2 |
| R27 | $HOCl + O = ClO + OH$ | $1.7 \times 10^{-13}$ | 2 |
| R28 | $ClNO_2 + OH = HOCl + NO_2$ | $2.4 \times 10^{-12}e^{-1250/T}$ | 2 |
| R29 | $ClONO_2 + O = ClO + NO_3$ | $3.6 \times 10^{-12}e^{-840/T}$ | 2 |
| R30 | $ClONO_2 + OH = HOCl + NO_3$ | $1.2 \times 10^{-12}e^{-330/T}$ | 2 |
| R31 | $HCl + O = Cl + OH$ | $1.0 \times 10^{-11}e^{-3300/T}$ | 2 |
| R32 | $Cl + ClNO = NO + Cl_2$ | $5.8 \times 10^{-11}e^{100/T}$ | 2 |
| R33 | $Cl + HOCl = OH + Cl_2$ | $3.4 \times 10^{-12}e^{-130/T}$ | 2 |
| R34 | $ClO \xrightarrow{hv} Cl + O$ | #1.0/<ClO> | 2 |
| R35 | $Cl + H_2O_2 = HCl + HO_2$ | $1.1 \times 10^{-11}e^{-980/T}$ | 2 |
| R36 | $OH + Cl_2 = HOCl + Cl$ | $2.6 \times 10^{-12}e^{-1100/T}$ | 2 |
| R37 | $Cl + HNO_3 = HCl + NO_2$ | $2.0 \times 10^{-16}$ | 2 |
| R38 | $Cl_2O_2 \xrightarrow{hv} Cl + ClO_2$ | #1.0/<Cl$_2$O$_2$> | 2 |
| R39 | $ClO_2 \xrightarrow{hv} ClO + O_2$ | #1.0/<ClO$_2$> | 2 |
| R40 | $Cl + ClO_2 = 2ClO$ | $1.2 \times 10^{-11}$ | 2 |
| R41 | $Cl + ClO_2 = Cl_2 + O_2$ | $2.3 \times 10^{-10}$ | 2 |
| R42 | $OH + Cl_2O_2 = HOCl + ClO_2$ | $6.0 \times 10^{-13}e^{670/T}$ | 2 |
| R43 | $OH + ClO_2 = HOCl + O_2$ | $1.4 \times 10^{-12}e^{600/T}$ | 2 |
| R44 | $Cl_2 + O_2 + M = Cl_2O_2$ | 2.2e-33^3.1&1.8e-10^0&-1.0&0.6&1.0 | 2 |
| R45 | $ClO + ClO + M = ClO_2$ | 1.9e-32^3.6&3.7e-12^1.6&-1.0&0.6&1.0 | 2 |
| R46 | $ClO_2 + O = ClO$ | $2.4 \times 10^{-12}e^{-960/T}$ | 2 |
| R47 | $NO + ClO_2 = ClO + NO_2$ | $6.0 \times 10^{-13}e^{-670/T}$ | 2 |
| R48 | $HCl + NO_3 = HNO_3 + Cl$ | $5.0 \times 10^{-17}$ | 2 |
| R49 | $Cl + Cl_2O_2 = Cl_2 + ClO$ | $6.2 \times 10^{-11}e^{130/T}$ | 2 |
| R50 | $ClO + O_3 = ClO_2$ | $2.0 \times 10^{-12}e^{-3600/T}$ | 2 |
| R51 | $ClO + NO_3 = ClO_2 + NO_2$ | $4.7 \times 10^{-13}$ | 2 |

1. Sander et al. (2010); 2. Burkholder et al. (2020)
**Table 2.** List of aqueous-phase chlorine and bromine reactions used in this study.

| No | Species | Reaction | Reaction rate (unit: M/s) | Reference |
|---|---|---|---|---|
| R1 | HOCl | $HOCl + HSO_3^- \xrightarrow{k} SO_4^{2-} + Cl^- + 2H^+$ | $2.8 \times 10^5$ | 1 |
| R2 | HOCl | $HOCl + SO_3^{2-} \xrightarrow{k} SO_4^{2-} + HCl$ | $7.6 \times 10^8$ | 1 |
| R3 | HOBr | $HOBr + HSO_3^- \xrightarrow{k} SO_4^{2-} + HBr$ | $5.0 \times 10^9$ | 1 |
| R4 | HOBr | $HOBr + SO_3^{2-} \xrightarrow{k} SO_4^{2-} + HBr$ | $2.6 \times 10^7$ | 1 |

1. Liu and Abbatt. (2020)

**Table 3**. List of heterogeneous halogen reactions and the uptake coefficients of gases used in
this study.

| No | Gas | Reaction | Uptake Coefficient ($\gamma_{gas}$) | Reference |
|---|---|---|---|---|
| R1 | $N_2O_5$ | $N_2O_5 + (1-\varphi)H_2O + \varphi Cl^- \rightarrow (2-\varphi)\ HNO_3 + \varphi ClNO_2$ | Function of aerosol mass concentrations, relative humidity, and temperature | 1, 2 |
| R2 | HOCl | $HOCl + Cl^- = Cl_2$ | $1.09 \times 10^{-3}$ | 3 |
| R3 | $ClONO_2$ | $ClONO_2 + Cl^- = Cl_2 + NO_3^-$ | 0.002 | 4 |
| R4 | $ClNO_2$ | $ClNO_2 + Cl^- + H^+ = Cl_2 + HONO$ | $2.65 \times 10^{-6}$ (pH < 2.0) | 5 |
| R5 | $ClNO_2$ | $ClNO_2 = Cl^- + NO_3^- + 2H^+$ | $6 \times 10^{-6}$ (pH > 2.0) | 6 |
| R6 | $NO_2$ | $2NO_2 + Cl^- = ClNO + NO_3^-$ | $10^{-4}$ | 7 |
| R7 | HOBr | $HOBr + Br^- = Br_2$ | 0.08 | 8 |
| R8 | HOBr | $HOBr + Cl^- = BrCl$ | 0.02 | 8 |
| R9 | $BrONO_2$ | $BrONO_2 + H_2O = HOBr + HNO_3$ | 0.03 | 9 |
| R10 | HBr | $HBr = Br^-$ | $1.3 \times 10^{-8} e^{\frac{4290}{T}}$ | 10 |
| R11 | $IONO_2$ | $IONO_2 + Cl^- = ICl + HNO_3$ | 0.005 | 11 |
| R12 | $IONO_2$ | $IONO_2 + Br^- = IBr + HNO_3$ | 0.005 | 11 |
| R13 | $INO_2$ | $INO_2 + Cl^- = ICl + HONO$ | 0.01 | 11 |
| R14 | $INO_2$ | $INO_2 + Br^- = IBr + HONO$ | 0.01 | 11 |
| R15 | HOI | $HOI + Cl^- = ICl$ | 0.005 | 12 |
| R16 | HOI | $HOI + Br^- = IBr$ | 0.005 | 12 |
| R17 | $I_2O_2$ | $I_2O_2 =$ | 0.02 | 12 |
| R18 | $I_2O_3$ | $I_2O_3 =$ | 0.02 | 12 |
| R19 | $I_2O_4$ | $I_2O_4 =$ | 0.02 | 12 |

1. Riemer et al. (2003); 2. Evans and Jacob (2005); 3.Pratte and Rossi (2006); 4.Chen et al. (2022); 5.Riedel et al.
(2012); 6.Roberts et al. (2009); 7.Abbatt and Waschewsky (1998); 8.Fernandez et al. (2014); 9.Deiber et al. (2004);
10.Ammann et al. (2013); 11.Saiz Lopez et al. (2014); 12.Sherwen et al. (2016)

## 2.4.2 Bromine reactions


We also incorporated bromine reactions into the SAPRC07-TC mechanism. The
reactions incorporated include: (i) updated absorption cross-sections for BrCl, BrCHO,
$CHBr_2Cl$, $CHBr_3$, and $CHBrCl_2$ (refer to R29 to R33 in Table 4); (ii) updated reaction rates for
formaldehyde (HCHO) and acetaldehyde ($CH_3CHO$) reacting with bromine radicals (refer to
R34 and R35 in Table 4); (iii) Br-initiated VOC reactions (refer to R36 to R56 in Table 4); (iv)
two inter-halogen species reactions (refer to R57 and R58 in Table 4); (v) two aqueous-phase
reactions (refer to R3 to R4 in Table 2); and (vi) four heterogeneous reactions for bromine
species (refer to R7 to R10 in Table 3). A complete list of the bromine reactions can be found
in Tables 2, 3, and 4.

**Table 4**. List of homogeneous bromine reactions used in this study.

| No | Reaction | Reaction rate | Reference |
|---|---|---|---|
| R1 | $BrO \xrightarrow{hv} Br + O^3P$ | #1.0/<BrO> | 1 |
| R2 | $HOBr \xrightarrow{hv} Br + OH$ | #1.0/<HOBr> | 1 |
| R3 | $BrNO_3 \xrightarrow{hv} Br + NO_3$ | #1.0/<BrNO_3\_1> | 1 |
| R4 | $BrNO_3 \xrightarrow{hv} BrO + NO_2$ | #1.0/<BrNO_3\_2> | 1 |
| R5 | $BrNO_2 \xrightarrow{hv} Br + NO_2$ | #1.0/<BrNO_2> | 1 |
| R6 | $Br_2 \xrightarrow{hv} 2Br$ | #1.0/<Br_2> | 1 |
| R7 | $Br + O_3 = BrO$ | $1.6 \times 10^{-11}e^{-780/T}$ | 1 |
| R8 | $BrO + HO_2 = HOBr$ | $4.5 \times 10^{-12}e^{460/T}$ | 1 |
| R9 | $Br + HO_2 = HBr$ | $4.8 \times 10^{-12}e^{-310/T}$ | 1 |
| R10 | $HBr + OH = Br$ | $6.7 \times 10^{-12}e^{155/T}$ | 1 |
| R11 | $BrO + BrO = 2Br$ | $1.4 \times 10^{-12}e^{210/T}$ | 1 |
| R12 | $BrO + BrO = Br_2$ | $2.9 \times 10^{-14}e^{840/T}$ | 1 |
| R13 | $BrO + NO = Br + NO_2$ | $8.8 \times 10^{-12}e^{260/T}$ | 1 |
| R14 | $Br + BrNO_3 = Br_2 + NO_3$ | $4.9 \times 10^{-11}$ | 1 |
| R15 | $Br_2 + OH = HOBr + Br$ | $2.1 \times 10^{-11}e^{240/T}$ | 1 |
| R16 | $BrO + OH = Br + HO_2$ | $1.7 \times 10^{-11}e^{250/T}$ | 1 |
| R17 | $Br + NO_3 = BrO + NO_2$ | $1.6 \times 10^{-11}$ | 1 |
| R18 | $BrO + ClO = Br + Cl$ | $4.7 \times 10^{-12}e^{320/T}$ | 1 |
| R19 | $BrO + MEO2 = 0.8HOBR + 0.2BR + 0.3HCOOH + 0.2HCHO + 0.13OH + 0.13HO_2$ | $2.65 \times 10^{-14}e^{1600/T}$ | 1 |
| R20 | $CH3Br + OH = Br$ | $1.42 \times 10^{-12}e^{-1150/T}$ | 1 |
| R21 | $MB3^a + OH = 3Br$ | $9.0 \times 10^{-13}e^{-360/T}$ | 1 |
| R22 | $MB2^b + OH = 2Br + HO_2$ | $2.0 \times 10^{-12}e^{-840/T}$ | 1 |
| R23 | $MB2C^c + OH = 2Br + Cl$ | $9.0 \times 10^{-13}e^{-420/T}$ | 1 |
| R24 | $MBC2^d + OH = Br + 2Cl$ | $9.4 \times 10^{-13}e^{-510/T}$ | 1 |
| R25 | $MBC2^e + OH = Br + Cl + HO_2$ | $2.1 \times 10^{-12}e^{-880/T}$ | 1 |
| R26 | $DMS + BrO = 0.75SO_2 + 0.25MSA + MEO2 + Br$ | $1.5 \times 10^{-14}e^{1000/T}$ | 1 |
| R27 | $BrO + NO_2 = BrNO_3$ | 5.2e-31^3.2&6.9e-12^-2.9&0.6&1.0 | 1 |
| R28 | $Br + NO_2 = BrNO_2$ | 4.2e-31^2.4&2.7e-11^0.0&0.6&1.0 | 1 |
| R29 | $BrCl \xrightarrow{hv} Br + Cl$ | #1.0/<BrCl> | 2 |
| R30 | $FMBR^f \xrightarrow{hv} Br + CO + HO2$ | #1.0/<FMBR> | 2 |
| R31 | $MB3^a \xrightarrow{hv} 3.0Br + HO_2$ | #1.0/<MB3> | 2 |
| R32 | $MB2C^c \xrightarrow{hv} 2.0Br + Cl + HO_2$ | #1.0/<MB2C> | 2 |
| R33 | $MBC2^d \xrightarrow{hv} Br + 2.0Cl + HO_2$ | #1.0/<MBC2> | 2 |
| R34 | $HCHO + Br = HBr + HO_2$ | $7.7 \times 10^{-12}e^{-580/T}$ | 2 |
| R35 | $CCHO + Br = HBr + MECO_3$ | $1.8 \times 10^{-11}e^{-460/T}$ | 2 |
| R36 | $ETHE + Br = FMBR^f + HCHO + HO2 + RO2C$ | $1.3 \times 10^{-13}$ | 2 |
| R37 | $OLE1 + Br = FMBR^f + CCHO + HO2 + RO2C$ | $3.6 \times 10^{-12}$ | 2 |
| R38 | $OLE2 + Br = FMBR^f + 0.75RCHO + 0.15ACET + 0.1MEK + HO2 + RO2C$ | $1.0 \times 10^{-11}$ | 2 |
| R39 | $ISOPRENE + Br = FMBR^f + PRD2 + HO2 + RO2C$ | $7.5 \times 10^{-11}$ | 2 |
| R40 | $FMBR^f + OH = Br + CO$ | $5.0 \times 10^{-12}$ | 2 |
| R41 | $RCHO + Br = HBr + RCO3$ | $1.8 \times 10^{-11}e^{-460/T}$ | 2 |
| R42 | $GLY + Br = HBr + 2CO + HO2$ | $7.7 \times 10^{-12}e^{-580/T}$ | 2 |
| R43 | $MGLY + Br = HBr + CO + MECO3$ | $1.8 \times 10^{-11}e^{-460/T}$ | 2 |
| R44 | $BALD + Br = HBr + BZCO3$ | $1.8 \times 10^{-11}e^{-460/T}$ | 2 |
| R45 | $ACROLEIN + Br = 0.75 FMBR^f + 0.25HBr + 0.25MACO3 + 0.75MGLY + 0.75HO2$ | $1.0 \times 10^{-11}$ | 2 |
| R46 | Benzene + Br = Product | $1.0 \times 10^{-11}$ | 3 |
| R47 | Toluene + Br = Product | $1.0 \times 10^{-14}$ | 4 |
| R48 | o-xylene + Br = Product | $1.6 \times 10^{-14}$ | 4 |
| R49 | m-xylene + Br = Product | $7.9 \times 10^{-14}$ | 4 |
| R50 | p-xylene + Br = Product | $4.0 \times 10^{-14}$ | 4 |
| R51 | $\alpha$-pinene + Br = Product | $6.8 \times 10^{-14}$ | 5 |
| R52 | MACR + Br = Product | $3.9 \times 10^{-10}$ | 5 |
| R53 | MVK + Br = Product | $2.3 \times 10^{-10}$ | 5 |
| R54 | MEK + Br = Product | $3.6 \times 10^{-10}$ | 5 |
| R55 | ETOH + Br = Product | $8.6 \times 10^{-11}e^{45/T}$ | 5 |
| R56 | MEOH + Br = Product | $5.5 \times 10^{-11}$ | 5 |
| R57 | $Cl + BrCl = Br + Cl_2$ | $1.45 \times 10^{-11}$ | 6 |
| R58 | $Cl + Br_2 = BrCl + Br$ | $1.94 \times 10^{-10}$ | 7 |

1. Sherwen et al. (2016); 2. Burkholder et al. (2020); 3. Keefer et al. (1950); 4. Giri et al. (2022); 5. Qinyi et al. (2021); 6. Clyne and Cruse. (1972); 7.Khamaganov and Crowley. (2010)

$MB3^a = CHBr_3$, $MB2^b = CH_2Br_2$, $MB2C^c = CH_2Br_2$, $MBC2^d = CHBr_2Cl$, $MBC^e = CH_2ClBr$, $FMBR^f = BrCHO$

### 2.4.3 Iodine reactions

Iodine reactions taken into account in this study were acquired from Saiz-Lopez et al. (2014) and Sherwen et al. (2016). We updated iodine reactions in three ways: (i) updated absorption cross-sections for ICl and IBr (refer to R43 and R44 in Table 5); (ii) two inter-halogen species reactions (refer to R45 and R46 in Table 5); and (iii) nine heterogeneous reactions for $IONO_2$, $INO_2$, HOI, $I_2O_2$, $I_2O_3$, and $I_2O_4$ (refer to R11 to R19 in Table 3). These iodine reactions are shown in Tables 3 and 5.

**Table 5.** List of homogeneous iodine reactions used in this study.

| No | Reaction | Reaction rate | Reference |
|---|---|---|---|
| R1 | $I_2 \xrightarrow{h\nu} 2I$ | #1.0/<I$_2$> | 1 |
| R2 | $HOI \xrightarrow{h\nu} I + OH$ | #1.0/<HOI> | 1 |
| R3 | $IO \xrightarrow{h\nu} I + O^3P$ | #1.0/<IO> | 1 |
| R4 | $OIO \xrightarrow{h\nu} I$ | #1.0/<OIO> | 1 |
| R5 | $INO \xrightarrow{h\nu} I + NO$ | #1.0/<INO> | 1 |
| R6 | $INO_2 \xrightarrow{h\nu} I + NO_2$ | #1.0/<INO$_2$> | 1 |
| R7 | $IONO_2 \xrightarrow{h\nu} I + NO_3$ | #1.0/<IONO$_2$> | 1 |
| R8 | $I_2O_2 \xrightarrow{h\nu} I + OIO$ | #1.0/<IONO$_2$> | 1 |
| R9 | $I_2O_3 \xrightarrow{h\nu} IO + OIO$ | #1.0/<IONO$_2$> | 1 |
| R10 | $I_2O_4 \xrightarrow{h\nu} 2OIO$ | #1.0/<IONO$_2$> | 1 |
| R11 | $CH_3I \xrightarrow{h\nu} I + MEO2$ | #1.0/<CH$_3$I> | 1 |
| R12 | $MIC \xrightarrow{h\nu} I + Cl$ | #1.0/<MIC> | 1 |
| R13 | $MIB \xrightarrow{h\nu} I + Br$ | #1.0/<INO$_2$> | 1 |
| R14 | $MI2 \xrightarrow{h\nu} 2I$ | #1.0/<IONO$_2$> | 1 |
| R15 | $I + O_3 = IO$ | $2.1 \times 10^{-11}e^{-830/T}$ | 2 |
| R16 | $I + HO_2 = HI$ | $1.5 \times 10^{-11}e^{-1090/T}$ | 2 |
| R17 | $I_2 + OH = HOI + I$ | $2.1 \times 10^{-10}$ | 2 |
| R18 | $HI + OH = I$ | $1.6 \times 10^{-11}e^{440/T}$ | 2 |
| R19 | $HOI + OH = IO$ | $5.0 \times 10^{-12}$ | 2 |
| R20 | $IO + HO_2 = HOI$ | $1.4 \times 10^{-11}e^{540/T}$ | 2 |
| R21 | $IO + NO = I + NO_2$ | $7.15 \times 10^{-12}e^{300/T}$ | 2 |
| R22 | $INO + INO = I_2 + 2NO$ | $8.4 \times 10^{-11}e^{-2620/T}$ | 2 |
| R23 | $INO_2 + INO_2 = I_2 + NO_2$ | $4.7 \times 10^{-13}e^{-1670/T}$ | 2 |
| R24 | $I_2 + NO_3 = I + IONO_2$ | $1.5 \times 10^{-12}$ | 2 |
| R25 | $IONO_2 + I = I_2 + NO_3$ | $9.1 \times 10^{-11}e^{-146/T}$ | 2 |
| R26 | $I + BrO = IO + Br$ | $1.2 \times 10^{-11}$ | 2 |
| R27 | $IO + Br = I + BrO$ | $2.7 \times 10^{-11}$ | 2 |
| R28 | $IO + BrO = Br + I$ | $1.5 \times 10^{-11}e^{510/T}$ | 2 |
| R29 | $IO + ClO = I + Cl$ | $4.7 \times 10^{-12}e^{280/T}$ | 2 |
| R30 | $OIO + OIO = I_2O_4$ | $1.5 \times 10^{-10}$ | 2 |
| R31 | $OIO + NO = IO + NO_2$ | $1.1 \times 10^{-12}e^{542/T}$ | 2 |
| R32 | $IO + IO = 0.4OIO + 0.4I + 0.6I_2O_2$ | $5.4 \times 10^{-11}e^{180/T}$ | 2 |
| R33 | $IO + OIO = I_2O_3$ | $1.5 \times 10^{-10}$ | 2 |
| R34 | $I_2O_2 = OIO + I$ | $2.5 \times 10^{-14}e^{-9770/T}$ | 2 |
| R35 | $I_2O_4 = 2OIO$ | $3.8 \times 10^{-2}$ | 2 |
| R36 | $INO_2 = NO_2 + I$ | $9.9 \times 10^{17}e^{-11859/T}$ | 2 |
| R37 | $IONO_2 = NO_2 + IO$ | $2.1 \times 10^{15}e^{-13670/T}$ | 2 |
| R38 | $CH_3I + OH = HCHO$ | $4.3 \times 10^{-12}e^{-1120/T}$ | 2 |
| R39 | $IO + DMS = 0.75SO_2 + 0.25MSA + MEO2$ | $3.3 \times 10^{-13}e^{-925/T}$ | 2 |
| R40 | $I + NO = INO$ | 1.8e-32^-1.0&1.7e-11^0.0&0.6&1.0 | 2 |
| R41 | $I + NO_2 = INO_2$ | 3.0e-31^-1.0&6.6e-11^0.0&0.6&1.0 | 2 |
| R42 | $IO + NO_2 = IONO_2$ | 7.7e-31^-3.5&7.7e-12^-.1.5&0.6&1.0 | 2 |
| R43 | $ICl \xrightarrow{h\nu} I + Cl$ | #1.0/<ICl> | 3 |
| R44 | $IBr \xrightarrow{h\nu} I + Br$ | #1.0/<IBr> | 3 |
| R45 | $Cl + I_2 = ICl + I$ | $2.81 \times 10^{-10}$ | 4 |
| R46 | $Br + I_2 = IBr + I$ | $1.2 \times 10^{-10}$ | 5 |

1. Sherwen et al. (2016); 2. Saiz-Lopez et al. (2014); 3. Burkholder et al. (2020); 4. Baklanov et al. (1997);
5. Bedjanian et al. (1997)

 **2.5 Experimental design**

To better understand the impacts of atmospheric halogen chemistry, we designed four

experiments: (i) experiment without halogen chemistry (referred to as CTRL); (ii) original
CMAQv5.2.1 model simulation only with chlorine processes ($EXP_{Cl}$); (iii) experiment with
both chlorine and bromine processes ($EXP_{Cl\_Br}$); and (iv) experiment with full halogen
processes ($EXP_{Cl\_Br\_I}$). The design of these four experiments is explained in Table 6.

In addition, in order to further analyze our results, we carried out two more

experiments: (i) CMAQ model runs with the halogen chemistry constructed by Saiz-Lopez et
al. (2014) (labeled as $EXP_{CAM}$); and (ii) CMAQ model run with the halogen chemistry
constructed by Sarwar et al. (2015) (labeled as $EXP_{CMAQ}$). The former halogen chemistry was
included in a global CTM named CAM-Chem, while the latter was in the CMAQ model. That
is why we labeled these two experiments, $EXP_{CAM}$ and $EXP_{CMAQ}$, respectively.

**Table 6.** Description of the four experiments conducted in this study.

| Experiment | Chlorine | | Bromine | | Iodine | |
|---|---|---|---|---|---|---|
| | Emission | Reaction | Emission | Reaction | Emission | Reaction |
| CTRL | - | - | - | - | - | - |
| $EXP_{Cl}$ | √ | √ | - | - | - | - |
| $EXP_{Cl\_Br}$ | √ | √ | √ | √ | - | - |
| $EXP_{Cl\_Br\_I}$ | √ | √ | √ | √ | √ | √ |


**3. Results & Discussions**

In this section, we discuss the accuracy of new halogen chemistry and processes

through the comparison between simulated and measured mixing ratios of halogen-containing
compounds during the period of KORUS-AQ campaign. We then analyze the experimental
results to evaluate the impacts of atmospheric halogen chemistry and processes on key-species
concentrations in the atmosphere.
**3.1 Model performances**
**3.1.1 Observed vs Modeled $ClNO_2$ mixing ratios**
To evaluate the model performances, we used the mixing ratios of $ClNO_2$ observed at
two supersites (Olympic Park and Mt. Taehwa stations) in South Korea. Although the mixing
ratios of atmospheric $Cl_2$ were also measured at these two stations, we focused solely on $ClNO_2$
observations due to several uncertainties associated with $Cl_2$ analysis. These issues will be
discussed later in Sect. 3.1.3.
Figure 2 presents the diurnal variations of modeled and observed mixing ratios of
$ClNO_2$ at two monitoring stations. The CTRL simulation (black circles and lines in Fig. 2) and
$EXP_{CAM}$ (purple circles and lines in Fig. 2) could not reproduce the observed mixing ratios of
$ClNO_2$ at the two supersites. For instance, the average mixing ratios of $ClNO_2$ at both
monitoring stations were 0.00 ppt for CTRL and 15.40 ppt for $EXP_{CAM}$, while the observed
average mixing ratio of $ClNO_2$ was 122.67 ppt. This large discrepancy may be primarily due
to the absence of heterogeneous $ClNO_2$ formation via $N_2O_5$ in the halogen scheme
implemented for $EXP_{CAM}$, which was based on Saiz-Lopez et al. (2014). It is noted that this
mechanism was originally designed for clean, oceanic environments and does not account for
anthropogenic chlorine sources or inland $ClNO_2$ formation pathways, such as the reactions of
$N_2O_5$ with particulate chloride (recall R8: $N_2O_5 + (1-\varphi)H_2O + \eta Cl^- \rightarrow (2-\varphi)HNO_3 +$
$\varphi ClNO_2$).
On the other hand, the $EXP_{CMAQ}$ (blue circles and lines in Fig. 2) and $EXP_{Cl\_Br\_I}$ (red
circles and lines in Fig. 2), which accounted for halogen chemistry, tend to better capture the
diurnal patterns of observed mixing ratios of $ClNO_2$ (refer to open circles and lines in Fig. 2).
These models also demonstrated significant improvements, in terms of statistical metrics
(which will be presented in Table 7).

Although the EXP$_{CMAQ}$ showed reasonable agreement, it also exhibited significant

biases during the nighttime. Conversely, the EXP$_{Cl\_Br\_I}$ simulation achieved better agreement
with the observed mixing ratios of ClNO$_2$ at both stations. For example, the index of agreement
(IOA) increased from 0.62 for EXP$_{CMAQ}$ to 0.66 for EXP$_{Cl\_Br\_I}$ and from 0.57 to 0.59 at
Olympic Park and Mt. Taehwa stations, respectively. These enhancements suggest that
successful implementation of the models depends on not only considering chlorine reactions
but also incorporating more comprehensive halogen reactions, as demonstrated by EXP$_{Cl\_Br\_I}$.
The following sections will explore and discuss these halogen reactions in the atmosphere.

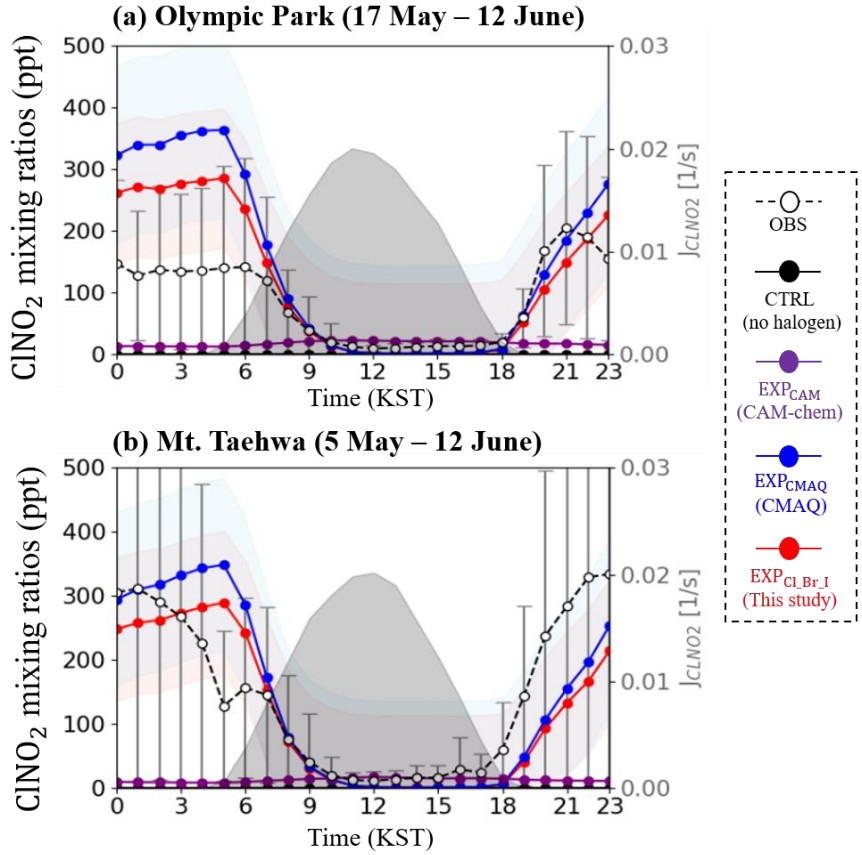


**Figure 2.** Diurnal variations in the mixing ratios of ClNO$_2$ (unit: ppt) at (a) Olympic Park and
(b) Mt.Taehwa stations during the period of the KORUS-AQ campaign. Observed values are
represented by open circles (error bars indicate the standard deviation). Colored lines with
shaded areas show the hourly-averaged mixing ratios of ClNO$_2$ and the corresponding standard

deviation from each simulation. The black shaded area indicates the variations in the photolysis
rate of $ClNO_2$ derived from the $EXP_{Cl\_Br\_I}$ simulation.
**Table 7.** Statistical metrics for $ClNO_2$ analysis from CTRL, $EXP_{CAM}$, $EXP_{CMAQ}$, and $EXP_{Cl\_Br\_I}$
simulations at the Olympic Park and Mt.Taewha stations during the period of KORUS-AQ
campaign.

| Target: $ClNO_2$ | | Olympic Park | Mt.Taehwa |
|---|---|---|---|
| | Observed mean (ppt) | 85.97 | 159.36 |
| CTRL | Modeled mean (ppt) | 0 | 0 |
| | MB (ppt) | -85.97 | -159.36 |
| | RMSE (ppt) | 150.30 | 288.82 |
| | IOA | 0.41 | 0.40 |
| $EXP_{CAM}$ | Modeled mean (ppt) | 17.31 | 13.49 |
| | MB (ppt) | -68.67 | -145.87 |
| | RMSE (ppt) | 143.23 | 281.78 |
| | IOA | 0.37 | 0.40 |
| $EXP_{CMAQ}$ | Modeled mean (ppt) | 135.77 | 152.89 |
| | MB (ppt) | 49.80 | -6.65 |
| | RMSE (ppt) | 204.42 | 289.19 |
| | IOA | 0.62 | 0.57 |
| $EXP_{Cl\_Br\_I}$ | Modeled mean (ppt) | 116.76 | 134.29 |
| | MB (ppt) | 30.79 | -25.07 |
| | RMSE (ppt) | 174.41 | 272.69 |
| | IOA | 0.66 | 0.59 |


### 3.1.2 Contributions to mixing ratios of $ClNO_2$

Figure 3a and 3c represent the diurnal variations in the mixing ratios of $ClNO_2$ from
the $EXP_{CMAQ}$ (blue dotted line) and $EXP_{Cl\_Br\_I}$ simulations (red dotted line) at two supersites
during the period of the KORUS-AQ campaign. Through a sensitivity test, we attempted to
identify the key reactions in the $EXP_{Cl\_Br\_I}$ causing differences from the $EXP_{CMAQ}$ simulation.
From the studies, we identified four critical halogen reactions: (i) updated reaction rate
coefficient of R20 in Table 1; (ii) newly added heterogeneous reaction of HOBr as shown in
R8 of Table 3; (iii) modified parameterization of $\gamma_{N2O5}$ as shown in R1 of Table 3; and (iv)
newly added heterogeneous reaction of $NO_2$ onto atmospheric aerosols as shown in R6 of Table
3. The contributions of these four reactions were calculated and are presented in blue-, green-,
yellow-, and red-shaded areas in Fig. 3, respectively. It is evident that the contribution of the
parameterization of $\gamma_{N2O5}$ is the most significant. A detailed analysis of these differences
between $EXP_{CMAQ}$ and the $EXP_{Cl\_Br\_I}$ is further discussed in Table S2.
Figure 3b and 3d illustrate the contributions of the four reactions to the mixing ratios
of $ClNO_2$ at the two supersites. Our results again indicate that selecting the new $\gamma_{N2O5}$ led to
the largest decreases in the averaged mixing ratios of $ClNO_2$ by 9.58 ppt (50.4%) and 7.50 ppt
(40.3%) at the Olympic Park and Mt. Taehwa stations, respectively. These reductions are
obviously attributed to lower values of $\gamma_{N2O5}$ in $EXP_{Cl\_Br\_I}$ ($\gamma_{N2O5}$ = ~0.013), compared to
those in $EXP_{CMAQ}$ ($\gamma_{N2O5}$ = ~0.014) as shown in Fig. S3. Such a small difference in $\gamma_{N2O5}$
led to substantial differences in the mixing ratios of $ClNO_2$. In addition, the inclusion of
reaction (recall R20: $ClO + ClO \rightarrow Cl_2$), with its rate constant reduced by a factor of 10 from
the original CMAQ model (see Table S2), led to a 3.40 ppt (17.8%) decrease in the $ClNO_2$
mixing ratio at the Olympic Park station. This effect is likely due to the slower removal of ClO,
which resulted in slightly elevated ClO levels in the boundary layer. The increased ClO may
have promoted the formation of $ClONO_2$ (i.e., reservoir species) via reaction with $NO_2$, thereby
reducing the amount of reactive nitrogen available for $ClNO_2$ production through the
heterogeneous pathway. Accounting for the heterogeneous reaction of $NO_2$ and HOBr onto
chlorine-containing particles also resulted in reductions in the mixing ratios of $ClNO_2$ by 4.46
ppt (24.0%) and 2.77 ppt (14.9%) at the Mt. Taehwa station, respectively. These reactions
competitively consume chloride-containing particles, thereby reducing the availability of
particulate chlorine that is essential for $ClNO_2$ formation. Overall, these results indicate that
the reduction in the $ClNO_2$ mixing ratios was primarily driven by the updated chlorine
chemistry, particularly the revised parameterization of $\gamma_{N2O5}$ and the inclusion of additional
chlorine-related reactions, with secondary contributions from bromine chemistry.
Collectively, the four reactions mentioned above may be the key reactions that can
significantly change the atmospheric levels of $ClNO_2$. Nevertheless, it should also be noted

that the $EXP_{Cl\_Br\_I}$ simulation still exhibited discrepancies with the observed mixing ratios of

$ClNO_2$. These remaining biases may be attributed to factors not fully accounted for in the

current modeling framework, including (i) the uptake of $Cl_2$ or related species onto aerosol

surfaces, (ii) uncertainties in the $ClNO_2$ yield, (iii) simplified diurnal emissions profiles for

HCl and $Cl_2$, and (iv) missing constrained halogen reactions. These limitations require further

targeted sensitivity analyses to better quantify their individual and combined impacts.

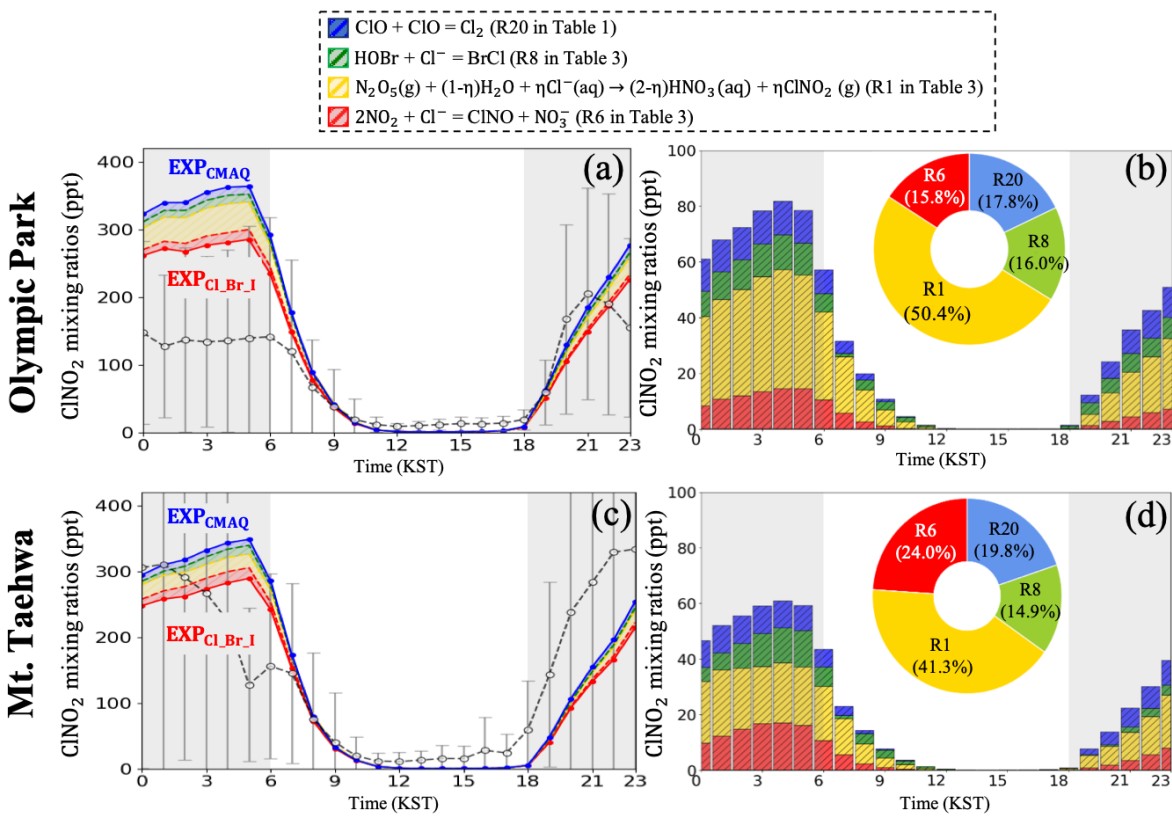

**Figure 3.** Contributions of halogen reactions to the mixing ratios of $ClNO_2$ in the $EXP_{CMAQ}$ and $EXP_{Cl\_Br\_I}$ simulations at (a and b) Olympic Park and (c and d) Mt. Taehwa stations during the period of KORUS-AQ campaign. Stacked bars and pie charts show the contributions from four halogen reactions to the mixing ratios of $ClNO_2$. Grey-shaded areas represent nighttime (18:00-06:00 local standard time).

### 3.1.3 Uncertainties in $Cl_2$

Figure 4 represents bar graphs of 24-hour averaged $Cl_2$ mixing ratios from the four

experiments, together with the observed $Cl_2$ mixing ratios at two supersites. Among them, the

mixing ratios of $Cl_2$ from the $EXP_{Cl\_Br\_I}$ agree well with the observed mixing ratios of $Cl_2$.

Based on these findings, we attempted to analyze which reactions contributed to elevated levels
of $Cl_2$. Two key reactions were identified: (i) $HOCl + Cl^- \rightarrow Cl_2$ (R2 in Table 3) and (ii)
$ClONO_2 + Cl^- \rightarrow Cl_2$ (R3 in Table 3). These reactions accounted for an increase in $Cl_2$ of
0.18 ppt (14.2%) and 1.06 ppt (84.1%), respectively, at the two supersites.

Although the current modeling system has improved the predictions of 'campaign-

averaged' $Cl_2$ mixing ratios, it has a serious limitation in reproducing daytime $Cl_2$ levels, likely
due to the extremely fast photo-dissociation rate of $Cl_2$ ($J_{Cl2}$ is estimated at $2.5 \times 10^{-3}$ s$^{-1}$). In
other words, once $Cl_2$ was produced via daytime halogen reaction pathways (see R3 to R5 in
Table 3), it was rapidly removed by the fast photo-dissociation. To address this challenge,
several studies suggested potential missing daytime reactions, such as particulate nitrate
photolysis and the uptake of $O_3$ and OH onto atmospheric particles (Peng et al., 2022; Chen et
al., 2022). However, these reactions also have limitations in perfectly explaining the relatively
high levels of $Cl_2$ during the daytime. Although the model slightly overestimates $Cl_2$ mixing
ratios during the nighttime (by approximately 0.3 ppt), the major discrepancy lies in the
inability to capture daytime peaks. In this context, the accuracy of simulating daytime $Cl_2$
mixing ratios remains a topic of further discussion.

In addition, significant uncertainties have been reported in observing $Cl_2$ mixing ratios

using the CIMS instrument. The detection limit for $Cl_2$ in the CIMS instrument was estimated
to be 2.9 ppt over a 30-minute interval (Jeong et al., 2019). However, the averaged mixing
ratios of $Cl_2$ of 2.08 ppt and 2.69 ppt were measured at Olympic Park and Mt. Taehwa,
respectively, as depicted in Fig. 4. Given that the observed levels of $Cl_2$ at the two monitoring
stations were very close to the detection limit of the instrument, significant uncertainties likely
exist in these measurements of the mixing ratios of $Cl_2$.

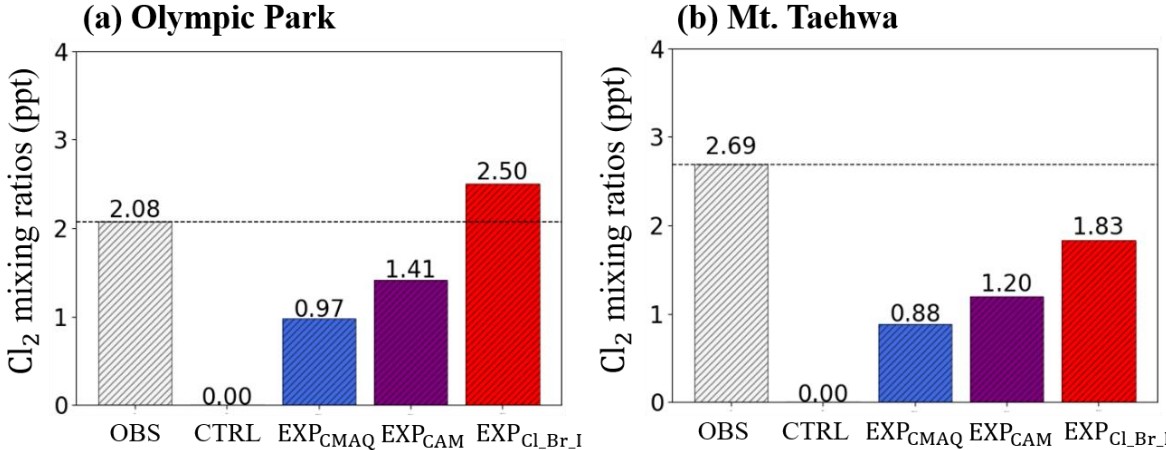

**Figure 4.** Comparisons of averaged mixing ratios of $Cl_2$ observed (OBS) and modeled from four simulations (CTRL, $EXP_{CMAQ}$, $EXP_{CAM}$, and $EXP_{Cl\_Br\_I}$) at (a) Olympic Park and (b) Mt. Taehwa stations. Dotted lines represent the observed mixing ratios of $Cl_2$.

### 3.1.4 Levels of bromine monoxide (BrO) and iodine monoxide (IO)

To strengthen the model evaluation, we also compared simulated bromine monoxide (BrO) and iodine monoxide (IO) levels with previously reported observational and modeling data, as shown in Fig. 5.

As illustrated in Fig. 5a, several modeling studies reported the mixing ratios of BrO ranging from 0.0 to 1.3 ppt (Koenig et al., 2017; Le Breton et al., 2017; Fan and Li, 2022; Zhu et al., 2019; Li et al., 2020; Li et al., 2019), which are similar to observed values (Koenig et al., 2017; Peters et al., 2005). Our simulation results indicate relatively low BrO mixing ratios, which may be attributed to low levels of chlorophyll-a over the Korean Peninsula during May and June (Son, 2013).

For IO, previous field and modeling studies have reported the mixing ratios of IO ranging from 0.0 to 2.5 ppt (Großmann et al., 2013; Allan et al., 2000; Fan and Li, 2022; Li et al., 2020; Takashima et al., 2021; Prados-Román et al., 2015; Mahajan et al., 2012; Mahajan et al., 2010; Inamdar et al., 2020). Overall, our simulated IO mixing ratios (0.0 – 1.8 ppt) over

the East Asian ocean region during the KORUS-AQ campaign is comparable to previously
reported ranges, as shown in Fig. 5b.

Although BrO and IO were not directly measured during the KORUS-AQ campaign,

this comparison suggests that our simulated halogen species are within a reasonable range of
previously reported values. These indirect evaluations complement the $ClNO_2$-based validation
and enhance confidence in our model's ability to simulate regional-scale behavior of bromine
and iodine species.

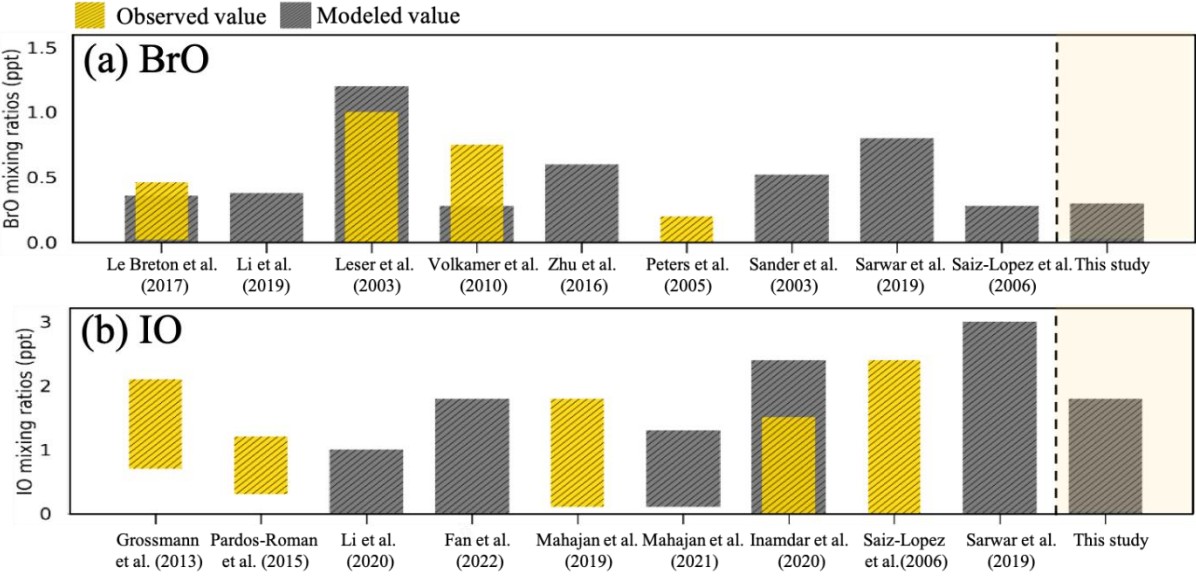


**Figure 5.** Comparisons between modeled and observed mixing ratios of atmospheric (a) BrO
and (b) IO. Both are observed and modeled mixing ratios were obtained from the previous
studies.

**3.2 Influences of halogen chemistry on $O_3$ mixing ratios**
**3.2.1 Comparative analysis at three supersites**

Based on the evaluation of model performances, we analyzed the impacts of halogen

processes on atmospheric $O_3$ mixing ratios at three monitoring stations (regarding the locations,
see Fig. 1a). Fig. 6 represents the diurnal variations in the mixing ratios of $O_3$ as simulated
from both the CTRL (black circles) and $EXP_{Cl\_Br\_I}$ (red circles), together with the observations
(white open circles) during the period of the KORUS-AQ campaign. It shows that the $O_3$
mixing ratios simulated from the $EXP_{Cl\_Br\_I}$ slightly increased by ~0.06 ppb (0.3%) and ~0.14
ppb (0.4%), higher than those from the CTRL at Olympic Park and Mt. Taehwa stations,
respectively. Similar patterns were also observed in the comparisons between $O_3$ observations
from 320 AIR-KOREA stations and $O_3$ predictions (as shown in Fig. S4).
It should be noted that the simulated $O_3$ mixing ratios decreased slightly by ~0.19 ppb
(0.3%), lower than those simulated from the CTRL at the Bangnyung station. These results
raised two questions: (i) why did the opposite patterns take place between two land stations
and one ocean station (Bangnyung Island station)? and (ii) what mechanism caused these
opposite patterns in the mixing ratios of $O_3$? To answer these two questions, we further
investigated the role of halogen chemistry in atmospheric $O_3$ chemistry.

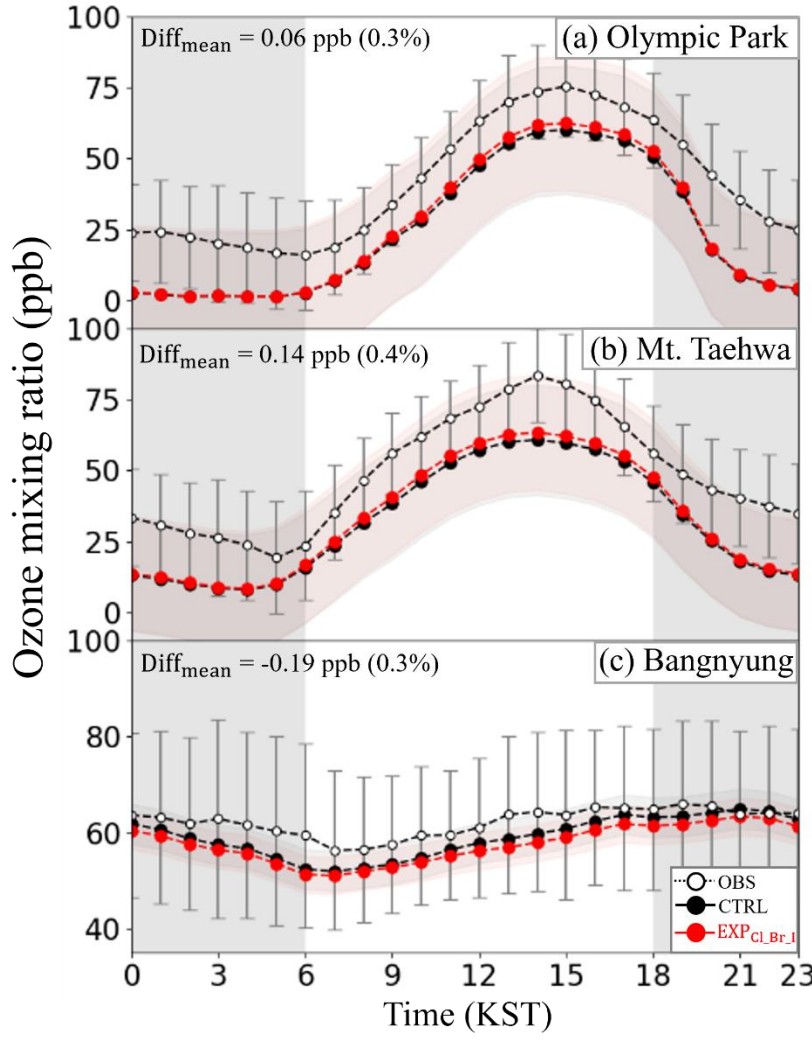


**Figure 6.** Diurnal variations in the mixing ratios of $O_3$ from CTRL (black circles) and
$EXP_{CL\_Br\_I}$ (red circles) simulations, together with observed mixing ratios of $O_3$ (OBS; white
circles) at (a) Olympic Park, (b) Mt. Taehwa, and (c) Bangnyung stations during the period of
the KORUS-AQ campaign. Error bars and shaded areas indicate the standard deviations of
observed and modeled $O_3$, while the grey-shaded areas show the nighttime. $DIFF_{mean}$ represents
the difference in the averaged mixing ratios of $O_3$ between $EXP_{Cl\_Br\_I}$ and CTRL simulations.

### 3.2.2 Impacts of halogen processes

Figure 7 illustrates the spatial distributions of the differences between two $O_3$ mixing
ratios simulated under the consideration of individual halogen chemistry over the Korean
peninsula. The CMAQ-simulated $O_3$ mixing ratios for $EXP_{Cl}$ showed an increase of 0.62 ppb
(1.4%) compared to those from the CTRL over the Korean peninsula (as shown in Fig. 7a).

This increase is attributed to the VOC oxidation by Cl radicals, which leads to the production
of additional $RO_2$ radicals. Additional $O_3$ is subsequently produced via the $RO_2$ + NO reactions.
It is well-known that VOC oxidation rates by Cl radicals are approximately 10 times faster than
those by OH radicals (Edwards and Young, 2024). Several previous studies have also
confirmed these findings (Kim et al., 2023; Jo et al., 2023).

Bromine-containing species are known to contribute to $O_3$ destruction in the

atmosphere. However, VOC oxidations by Br radicals also contribute to atmospheric $O_3$
formation via the reactions of Br + VOC $\rightarrow$ products + $RO_2$ (these reactions are shown in R36
to R57 in Table 4). As a result, the net effects of bromine processes (i.e., $EXP_{Cl\_Br} - EXP_{Cl}$)
lead to a slight increase in $O_3$ mixing ratios of ~0.01 ppb, as shown in Fig. 7b. This negligible
increase is likely due to the competition between $O_3$ loss via bromine-catalyzed destruction
and $O_3$ production via VOC oxidation.

In Fig. 7c, when iodine processes were incorporated into the modeling system, the

surface-averaged $O_3$ mixing ratios decreased by ~1.39 ppb (2.4%), particularly over ocean
areas. The iodine radicals generated from the photolysis of marine-originated iodine species
primarily react with $O_3$. Given the low levels of VOCs over ocean areas, iodine radicals
predominantly participate in the $O_3$ destruction over the ocean.

Again, the $O_3$ mixing ratios are controlled by competition between the $O_3$ production

and $O_3$ destruction. We found that the average $O_3$ mixing ratios increased by 0.21 ppb (~0.5%)
over land areas and decreased by 0.69 ppb (~1.2%) over ocean areas under the considerations
of entire halogen processes (i.e., $EXP_{Cl\_Br\_I} - CTRL$) (refer to Figs. 7d and S5). These findings
are closely in line with the increases in $O_3$ mixing ratios at the Olympic Park and Mt. Taehwa
stations (located on land areas) and the decreases in $O_3$ mixing ratios at the Bangnyung station
(located around ocean areas) under the comprehensive considerations of the halogen chemistry,
as discussed in Sect. 3.2.1. Such contrasting effects of halogen chemistry between polluted
continental and pristine oceanic regions are consistent with previous studies (e.g., Li et al.,
2022; Saiz-Lopez et al., 2023), which have also reported $O_3$ enhancements over land and
reductions over the ocean.

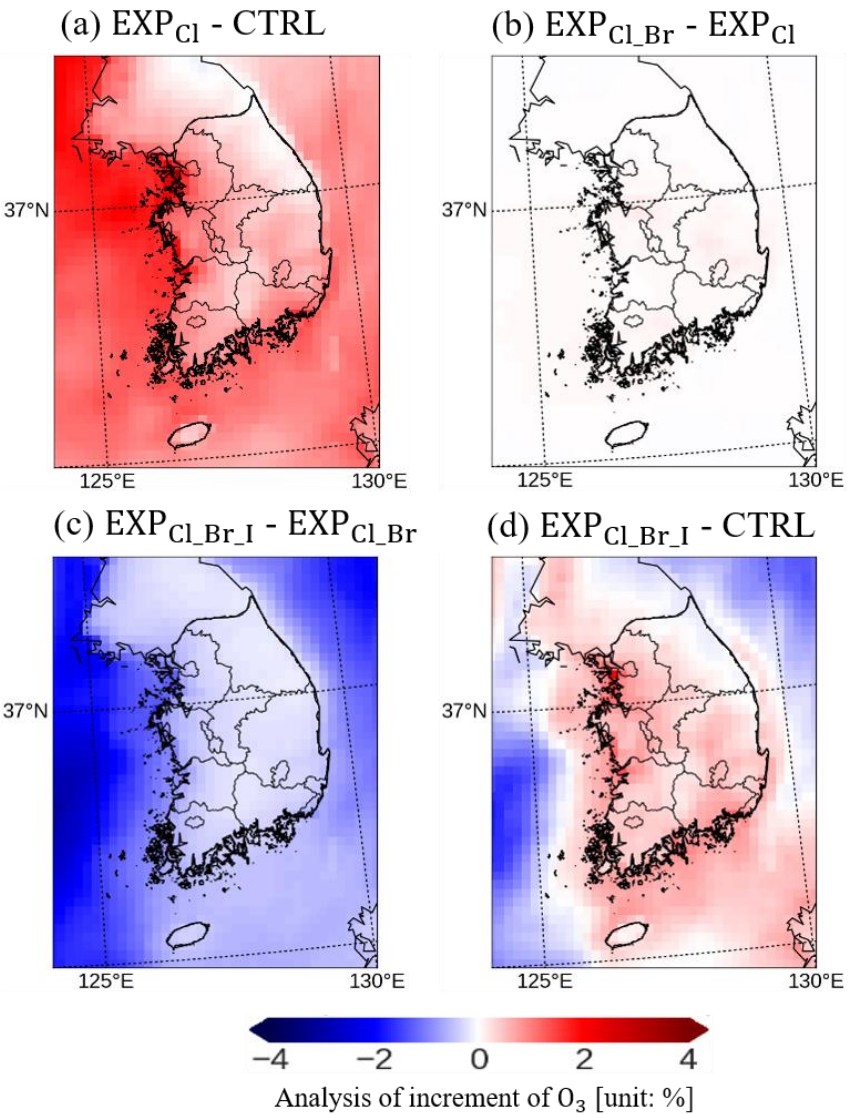


**Figure 7.** Spatial impacts of (a) chlorine processes ($EXP_{Cl}$ - CTRL), (b) bromine processes ($EXP_{Cl\_Br}$ – $EXP_{Cl}$), (c) iodine processes ($EXP_{Cl\_Br\_I}$ – $EXP_{Cl\_Br}$), and (d) total halogen processes ($EXP_{Cl\_Br\_I}$ – CTRL) on $O_3$ mixing ratios over the Korean peninsula.

### 3.2.3 Net $O_x$ production

To better understand the influences of the halogen chemistry on atmospheric $O_3$ mixing

ratios, we additionally carried out a quantitative analysis. We calculated $O_x$ production rates
(P(O$_x$)) utilizing a definition of expanded O$_x$ family ($\equiv$ O$_3$ + O$^{1D}$ + O$^{3p}$ + NO$_2$ + 2NO$_3$ +
3N$_2$O$_5$ + XNO$_2$ + XO + XONO$_2$; here, X denotes Cl, Br, and I). The constructions of the P(O$_x$)
are shown in equations (8) to (10):

$$F(O_x) = k_{HO_2+NO}[HO_2][NO] + k_{RO_2+NO}[RO_2][NO] \tag{Eq. 8}$$

$$D(O_x) = k_{NO_2+OH}[NO_2][OH] + k_{O_3+VOC}[O_3][VOC] + k_{O(1D)+H_2O}[O(^1D)][H_2O]$$

$$+ k_{O_3+OH}[O_3][OH] + k_{O_3+HO_2}[O_3][HO_2] + k_{RO_2+NO_2}[RO_2][NO_2]$$

$$+ (k_{XO+O_3}[O_3] + k_{XO+HO_2}[HO_2] + k_{XO+O(3p)}[O(^3P)])[XO]$$

$$+ k_{ClO+NO_2}[NO_2][ClO] + k_{XO+XO}[XO]^2 + k_{NO_3+VOC}[NO_3][VOC]$$

$$+ 3k_{het}[N_2O_5] \tag{Eq. 9}$$

$$P(O_x) = F(O_x) - D(O_x) \tag{Eq. 10}$$

where F(O$_x$) and D(O$_x$) represent the O$_x$ formation rates and O$_x$ destruction rates, respectively.
$k_i$ and $k_{het}$ denote the reaction rate constants for reaction $i$ and heterogeneous reactions of N$_2$O$_5$,
respectively.
Figure 8 shows the average O$_x$ formation rates (F(O$_x$)), O$_x$ destruction rates (D(O$_x$)),
and the P(O$_x$) from the CTRL and EXP$_{Cl\_Br\_I}$ simulations. Over land, rates for reactions of RO$_2$
+ NO and HO$_2$ + NO in the F(O$_x$) increased from 2.16 ppb/h to 2.21 ppb/h and from 2.05 ppb/h
to 2.09 ppb/h, respectively, with the full consideration of the halogen processes in the CMAQ
model. On the contrary, D(O$_x$) decreased from 1.12 ppb/h to 0.97 ppb/h due to the limited
contribution of O$_3$ destruction by halogen radicals. Consequently, P(O$_x$) increased from 3.08
ppb/h to 3.33 ppb/h. The enhanced F(O$_x$) of 0.09 ppb/h with decreased D(O$_x$) of 0.15 ppb/h
contributes to the increase in P(O$_x$) of 0.25 ppb/h. Consistent results were observed at the
Olympic Park and Mt. Taehwa stations (as shown in Figs. 6a and 6b).
On the other hand, the F(O$_x$) decreased by 0.03 ppb/h, while both D(O$_x$) and P(O$_x$)
increased by 0.12 ppb/h and 0.14 ppb/h, respectively, over the ocean (refer to Fig. 8b and Table
8). This may be caused by the halogen-related losses in $D(O_x)$ (caused mainly by $IO + HO_2$
reaction), significantly contributing to the $O_3$ destruction. These results indicate that the mixing
ratios of $O_3$ tend to decrease in the presence of iodine radicals over the ocean areas. This is also
in line with the case of the Bangnyung station, shown in Fig. 6c.

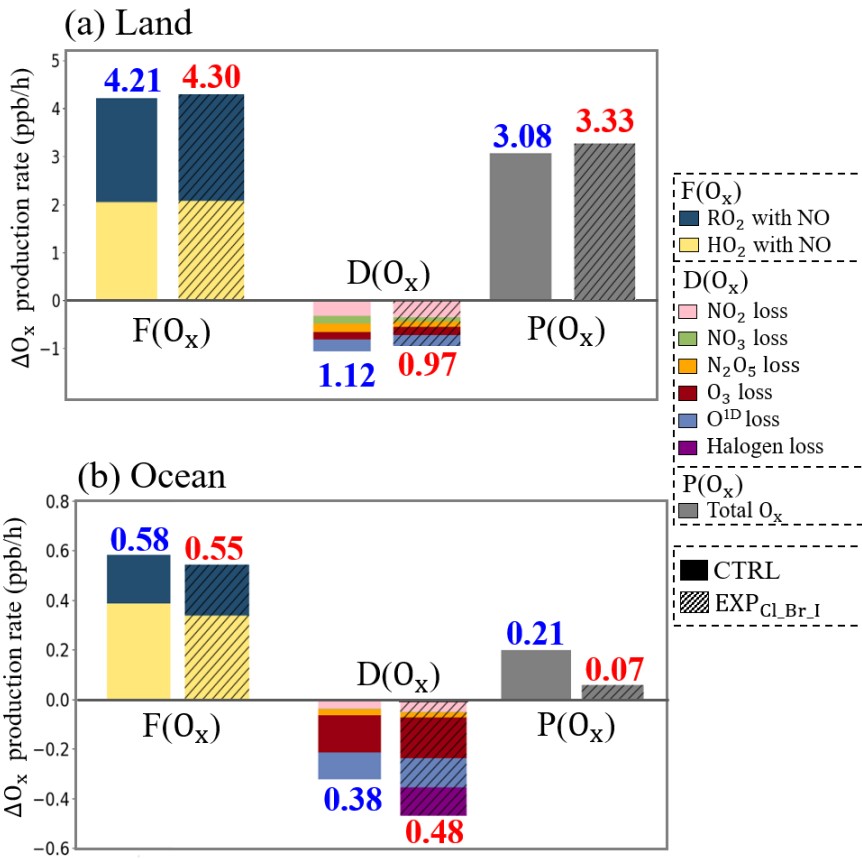


**Figure 8.** The stacked bar graphs represent $O_x$ formation rate ($F(O_x)$), destruction rate ($D(Ox)$),
and production rate ($P(O_x)$) in the CTRL (plain bars) and $EXP_{Cl\_Br\_I}$ (hatched bars) simulations
over the (a) land and (b) ocean areas, respectively, during the period of KORUS-AQ campaign.
Individual reactions contributing to $F(O_x)$ and $D(O_x)$ are indicated by the bar colors.

**Table 8.** Averaged budget for $O_x$ formation rates ($F(O_x)$), $O_x$ destruction rates ($D(O_x)$), and $O_x$ production rates ($P(O_x)$) calculated from the CTRL and $EXP_{Cl\_Br\_I}$ simulations during the period of KORUS-AQ campaign.

| | Land | | Ocean | |
|---|---|---|---|---|
| | CTRL | $EXP_{Cl\_Br\_I}$ | CTRL | $EXP_{Cl\_Br\_I}$ |
| **$O_x$ Formation: $F(O_x)$** | | | | |
| $RO_2 + NO$ | 2.16 | 2.21 | 0.20 | 0.21 |
| $HO_2 + NO$ | 2.05 | 2.09 | 0.39 | 0.34 |
| Total $F(O_x)$ | 4.21 | 4.30 | 0.58 | 0.55 |
| **$O_x$ Destruction: $D(O_x)$** | | | | |
| $NO_2$ loss[1] | 0.33 | 0.35 | 0.05 | 0.04 |
| $NO_3$ loss[2] | 0.17 | 0.09 | < 0.01 | < 0.01 |
| $N_2O_5$ loss[3] | 0.20 | 0.12 | 0.03 | 0.02 |
| $O_3$ loss[4] | 0.26 | 0.24 | 0.13 | 0.12 |
| $O^{1D}$ loss[5] | 0.16 | 0.17 | 0.18 | 0.17 |
| Halogen loss[6] | 0 | 0.01 | 0 | 0.12 |
| Total $D(O_x)$ | 1.12 | 0.97 | 0.38 | 0.48 |
| **Total $O_x$ production: $P(O_x)$** | | | | |
| $P(O_x) = F(O_x) - D(O_x)$ | 3.08 | 3.33 | 0.21 | 0.07 |

$NO_2$ loss[1]$= NO_2+RO_2$ and $NO_2+OH$
$NO_3$ loss[2]$= NO_3+VOC$
$N_2O_5$ loss[3]$=$ Heterogeneous reaction of $N_2O_5$
$O_3$ loss[4]$= O_3 + OH$, $O_3 + VOC$, and $O_3 + HO_2$
$O^{1D}$ loss[5]$= O^{1D}+H_2O$
Halogen loss[6]$= XO + O_3$, $HO_2$, and $O^{3p}$; $ClO + NO_2$; $XO + XO$ (where X denotes Cl, Br, and I)

## 3.3 Impacts of halogen chemistry on atmospheric species

Figure 9 summarizes the impacts of halogen processes on the mixing ratios of key atmospheric species over/around the Korean peninsula. The mixing ratios of hydroxyl radicals (OH) and hydroperoxyl radicals (HO2) increased by ~0.002 ppt (2.2%) and ~0.13 ppt (1.6%), respectively, when including the chlorine processes (i.e., $EXP_{Cl}$ - CTRL). This is due to the fact that the chlorine radicals in the atmosphere substitute the role of OH radicals. In other words, chlorine radicals actively react with VOCs. Thus, the VOC mixing ratios decreased by 0.39 ppb (1.0%) due to the active VOC oxidation by Cl radicals. Formaldehyde (HCHO), an intermediate product of VOC oxidation, increased by 0.02 ppb (1.1%) due to the enhanced rates of the VOC oxidation by Cl radicals. The $NO_x$ mixing ratios also increased by ~0.26 ppb

(2.6%). Higher levels of $NO_x$ may be due to the elevated levels of $ClNO_2$, which is a precursor
of $NO_x$ in the atmosphere.

We also explored the impacts of bromine processes (i.e., $EXP_{Cl\_Br} - EXP_{Cl}$). The

mixing ratios of OH, $HO_2$, and HCHO further increased by ~0.001 ppt (1.1%), ~0.03 ppt (0.4%),
and ~0.01 ppb (0.5%), respectively. These patterns appear to be similar to those in the chlorine
case.

The effects of iodine chemistry (i.e., $EXP_{Cl\_Br\_I} - EXP_{Cl\_Br}$) revealed that OH mixing

ratios increased by 0.004 ppt (3.1%). However, the mixing ratios of $HO_2$ decreased by 0.32 ppt
(3.8%). Interestingly, this increase in OH contrasts with a previous global-scale study (e.g., Li
et al., 2022), where iodine chemistry typically leads to lower OH due to halogen-driven $O_3$
suppression and reduced $O(^1D)$ production. This discrepancy can likely be attributed to
differences in the spatial and temporal scales of the analysis. Specifically, our study focuses on
a short-term episode (i.e., the KORUS-AQ campaign), during which OH mixing ratios may be
more strongly influenced by halogen-mediated radical regeneration. For instance, reactions
such as $IO + HO_2 \rightarrow HOI$, followed by HOI photolysis ($HOI \xrightarrow{h\nu} I + OH$), can provide an
additional OH source that partially compensate for the $O_3$-related OH loss. Similar findings
have been reported in previous studies (Saiz-Lopez et al., 2012; Mahajan et al., 2021; Stone et
al., 2018). The levels of HCHO and VOCs remain almost unchanged due to the fact that iodine
radicals do not strongly participate in the reactions with VOCs. The $NO_x$ levels increased
slightly by ~0.03 ppb (0.3%) on land and decreased by ~0.03 ppb (4.8%) over ocean areas,
which is in line with findings from the previous study (Mahajan et al., 2021).

Collectively, the influence of the full halogen chemistry (i.e., $EXP_{Cl\_Br\_I} - CTRL$)

shows that the OH mixing ratios increased significantly by 0.007 ppt (5.5%), while the $HO_2$
mixing ratios decreased by 0.45 ppt (5.3%) over ocean areas. These patterns are comparable in
magnitude to those reported in previous studies. For example, Stone et al. (2018) reported a 2%
increase in OH and a 5% decrease in $HO_2$, while Chen et al. (2024) found larger changes with
a 12% increase in OH and an 8% decrease in $HO_2$. In addition, Sarwar et al. (2015) found a
slight decrease in OH (1%) accompanied by a more pronounced decrease in $HO_2$ (11%). The
mixing ratios of HCHO and $NO_x$ increased by ~0.03 ppb (1.6%) and ~0.29 ppb (2.9%) over
the land. On the contrary, the mixing ratios of VOCs and $NO_x$ decreased by ~0.71 ppb (5.9%)
and ~0.05 ppb (7.8%) over the ocean areas, respectively. The reduction in $NO_x$ over ocean is
consistent with Wang et al. (2021), who reported a similar decrease of approximately 6% due
to halogen chemistry.
In addition, elevated oxidant capacity in the simulation of $EXP_{Cl\_Br\_I}$ results in
enhancements in the concentrations of sulfate by 0.05 $\mu g \cdot m^{-3}$ (1.5%) and secondary organic
aerosols by 0.14 $\mu g \cdot m^{-3}$ (1.8%). However, concentrations of nitrate and ammonium decreased
by ~1.60 $\mu g \cdot m^{-3}$ (29.4%) and ~0.55 $\mu g \cdot m^{-3}$ (5.0%), respectively, as shown in Fig. S6. This
resulted from using smaller uptake coefficient for $N_2O_5$ (refer to Fig. S3), which suppresses
$NH_4NO_3$ formation. As a result, $PM_{2.5}$ levels decreased from 21.59 $\mu g \cdot m^{-3}$ to 20.63 $\mu g \cdot m^{-3}$
(10.5%) over the Korean Peninsula during the KORUS-AQ campaign.

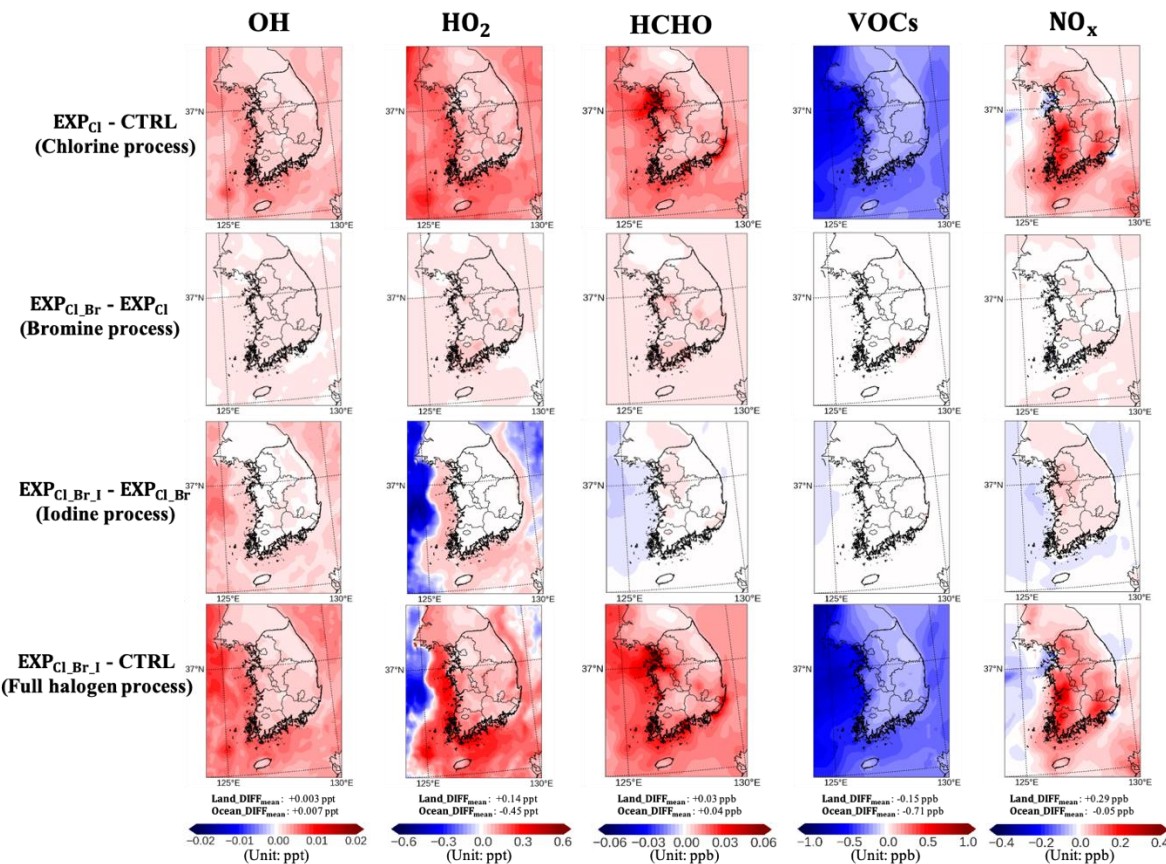

**Figure 9.** Summaries of the impacts of chlorine processes ($EXP_{Cl}$ – CTRL), bromine processes ($EXP_{Cl\_Br}$ – $EXP_{Cl}$), iodine processes ($EXP_{Cl\_Br\_I}$ – $EXP_{Cl\_Br}$), and full halogen processes ($EXP_{Cl\_Br\_I}$ – CTRL) on the mixing ratios of OH, $HO_2$, HCHO, VOCs, and $NO_x$, respectively, during the period of KORUS-AQ campaign. Also, Land_DIFF$_{mean}$ and Ocean_DIFF$_{mean}$ indicate the differences in the averaged mixing ratios of each species between $EXP_{Cl\_Br\_I}$ and CTRL simulations, over land and ocean, respectively.

## 4. Summary and Conclusions

To investigate the impacts of halogen chemistry over the Korean peninsula, we attempted to add and update reactions involving three halogen species (Cl, Br, and I) in the CMAQ modeling system. First, we estimated anthropogenic emissions of HCl, $Cl_2$, HBr, and $Br_2$ from five main sectors (such as industry, residential areas, power plants, solid waste incineration, and others). The anthropogenic emissions for HCl, $Cl_2$, HBr, and $Br_2$ were estimated to be 5,989.6, 450.8, 460.8, and 240.8 $Mg \cdot yr^{-1}$ over our research domain, respectively. Second, we also estimated emissions of natural halocarbons and inorganic bromine and iodine

($Br_2$, $I_2$, and HOI), based on the information derived from the GOCI sensor. Finally, we
embedded halogen chemical reactions (58 chlorine reactions, 64 bromine reactions, and 55
iodine reactions) into the CMAQ model.
We then tested the model performances in terms of the mixing ratios of $ClNO_2$ during
the period of the KORUS-AQ campaign at two supersites in South Korea. The $EXP_{Cl\_Br\_I}$
simulation exhibited the best performance in terms of the mixing ratios of $ClNO_2$. With the
$EXP_{Cl\_Br\_I}$ simulation, the IOA increased from 0.41 to 0.66 at the Olympic Park station, and
0.40 to 0.59 at the Mt. Taehwa station. Meanwhile, the MB decreased from -85.97 ppt to 30.79
ppt at the Olympic Park station, and -159.36 ppt to -25.07 ppt at the Mt. Taehwa station. This
is because the four following halogen reactions considered in this study contributed to better
$ClNO_2$ simulations: (i) $ClO + ClO \rightarrow Cl_2$; (ii) $HOBr + Cl^- \rightarrow BrCl$; (iii) different
parameterization of $\gamma_{N2O5}$; and (iv) $2NO_2 + Cl^- \rightarrow ClNO + NO_3^-$.
In addition to the evaluation of $ClNO_2$, we assessed the overall performance of the
implemented halogen chemistry by comparing simulated BrO and IO mixing ratios with
previously reported observational and modeling results. The simulated mixing ratios of BrO
and IO fall within the reported ranges of 0.0 – 1.3 ppt for BrO and 0.0 – 2.5 ppt for IO. This
indicates that the updated halogen processes can reasonably reproduce the regional-scale
behavior of bromine and iodine species.
Our study further emphasized the significant influences of individual halogen
processes on $O_3$ mixing ratios over South Korea. The average mixing ratios of $O_3$ increased by
~ 0.21 ppb (0.5%) over land areas due to the impacts of chlorine and bromine processes. On
the contrary, the $O_3$ mixing ratios decreased by ~0.69 ppb (1.2%) over ocean areas due to iodine
processes. In addition, we quantitatively calculated the $O_x$ budget. The net $O_x$ production rate
($P(O_x)$) increased from 3.08 to 3.33 ppb/h over the land areas and decreased from 0.21 to 0.07
ppb/h over the ocean areas with the simulation of the $EXP_{Cl\_Br\_I}$.
Finally, we further explored the impacts of full halogen processes on the atmospheric
composition. Compared with the CTRL simulation, the mixing ratios of HCHO and $NO_x$
increased by ~0.03 ppb (1.6%) and ~0.29 ppb (2.9%) over the land, respectively. On the other
hand, the mixing ratios of $HO_2$ and VOCs decreased by ~0.45 ppt (5.3%) and ~0.71 ppb (5.9%)
over the ocean areas, respectively, during the period of the KORUS-AQ campaign.
In conclusion, we believe that we successfully incorporated comprehensive halogen
processes into the CMAQ modeling system. Although our evaluation was limited to a few
halogen-containing species, the developed framework allowed us to explore the broader impact
of halogen chemistry on atmospheric composition and air quality. Despite these contributions,
several limitations in modeling fields remain, including (i) the spatiotemporal variability of
halogen emissions, (ii) incomplete or uncertain chemical mechanisms, and (iii) limited
observational data. In this context, further research is needed to better understand and reduce
these uncertainties.

## Code and data availability

After user registration, the WRF model 3.8.1 (https://doi.org/10.5065/D6MK6B4K, WRF User Page, 2024) and CMAQ v5.2.1 (https://doi.org/10.5281/zenodo.1079909, US EPA Office of Research and Development, 2015) are available from web page. The observation data we used can be accessed at https://www-air.larc.nasa.gov/cgi-bin/ArcView/korusaq?GROUND-NIER-OLYMPIC-PARK=1 (NASA, 2019).

## Author contributions
Conceptualization: KK, CHS, and KMH. Writing: KK, CHS, and KMH. Experimental design: KK, CHS, KMH, GY, and RB. Supervision: CHS. Validation: KK and CHS. Analysis: KK. Data curation: SK. All authors contributed to this paper for publication.

## Competing interests
At least one of the (co-)authors is a member of the editorial board of *Atmospheric Chemistry and Physics*. The authors declare that they have no known competing financial interests or personal relationships that could have appeared to influence the work reported in this paper.

## Acknowledgment
This work was supported by the National Research Foundation of Korea (NRF) grand funded by the Korea government (MSIT) (grant number: 2021R1A2C1006660).

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
