# Peer review of "Incorporation of multi-phase halogen chemistry into Community Multiscale Air Quality (CMAQ) model Kiyeon Kim1, Chul Han Song1\*, Kyung Man Han1, Greg Yarwood2, Ross Beardsley2, and Saewung Kim3 1. School of Earth Science and Environmental Engineering, Gwangju Institute of Science and Technology (GIST), Gwangju 61005, Republic of Korea 2. Ramboll, Novato, CA 94945, USA 3. Department of Earth System"

_EGUsphere, 2025_

## Referee Comment (RC2)

**Review of the manuscript "Incorporation of multi-phase halogen chemistry into Community Multiscale Air Quality (CMAQ) model" by Kim et al., EGUsphere preprint repository, 2025.**

The paper presents the implementation of halogen chemistry in the CMAQ model, including a comprehensive representation of chlorine, bromine and iodine sources and chemistry. Once the model updates are described, the study focus on reproducing ClNO2 and O3 observations over the Korean Peninsula during the KORUS-AQ campaign. The results focus on the improvements in the model-observation comparisons, including the index of agreement and other statistical parameters, particularly over two inland locations within a polluted / semi-polluted environment. Then, they identify the four main reactions in the model accounting for the largest fraction of the improvement. Based on this, the authors evaluate the influence of air quality in the whole modeled domain, highlighting the major and sometimes opposite differences observed over continental and oceanic domains, and provide general conclusions about the benefits of considering halogen chemistry in the study.

First of all, I would like to recognize the efforts from the group to implement their own version of halogen chemistry in CMAQ, which is of major importance as the community needs more modeling studies focused on the halogen influence on atmospheric chemistry and climate. However, I believe the current version of the work does not allow reaching a firm conclusion of the results obtained, probably because the paper attempts to address all at once the complete technical implementation, the observational improvements achieved over continental locations, and the overall impact and influences over different regions. While I leave for the authors to decide if it is convenient to present all of these developments within a single work or to partially split into independent companion papers, I recommend the authors to address the following major comments and submit a revised version for further consideration.

**Major Comments:**

P4,L93: In the introduction you clearly state that "We then investigate the formation of ozone using the new halogen processes", which is what the paper focus first … but then at the end the work attempts to provide much wider conclusions of the halogen influence on atmospheric chemistry and air quality. You may want to focus here on ozone production driven by ClNO2 chemistry, and leave the more general discussion for a companion paper.

P5,L111: Comparison with ClNO2 observations are of mayor importance for this study. Indeed, it might be worth to mention in the title. Current title and abstract give the impression of a general halogen chemistry development, while the work mostly focus on ClNO2 and its role for ozone production.

P20,L326-328: Your results for EXP_CAM are not surprising to me as the heterogeneous ClNO2 formation through N2O5 was not considered in Saiz-Lopez et al., (2014), which is your reference. However, it should be mentioned that those studies focused on oceanic and pristine conditions, and not polluted areas with high NOx and inland chlorine emissions. Indeed, further research from the group lead by Dr. Saiz-Lopez considered enhanced HCl and ClNO2 production within continental areas (see for example Li et al. 2022), which clearly showed important implications when anthropogenic chlorine sources are considered. Therefore I recommend including the complete chlorine scheme from CAM-Chem in your analysis to avoid reaching erroneous conclusion, or at least to clarify why your EXP_CAM simulations do not reproduce ClNO2 observations.

P22,L356-367: Of all four dominant process mentioned here, R20 in Table 1 (ClO + ClO) is particularly surprising to me, as this reaction is typically considered in stratospheric ozone depletion, but not for boundary layer studies. How do you explain that the surface observations are sensible to this reaction? Is it because of the subsequent Cl2 photolysis? Is due to the coupling of ClO + NO2? Can you explain how this reaction can contribute to ClNO2 formation during the night, when ClO abundance is zero and in addition, any Cl2 formed would not be photolyzed until the next morning?

P33,L547-562: I follow the explanation about the changes in the OH/HO2 partitioning, but oceanic SLH (particularly iodine) has been clearly shown to reduce the OH abundance (not increase it), and consequently to increase the CH4 burden and lifetime (Li et al., 2022). Your results over the ocean seems to contradict that. How could this be? Indeed, the null cycle mentioned in your work results in a shift of partitioning from OH to HO2 (which is fine), but the total OH abundance is controlled by O3 + hv --> O1D. Given that oceanic halogens reduce O3, there is less O1D and therefore OH formation should decrease. Have you discarded any influence from the BC affecting the overall results? Note that O3 changes for the EXP_Cl_Br_I - EXP_ctrl are in line with Li et al. (2022) for oceanic domains, but the OH changes are not consistent with the changes in O3 (unless I missed something).

**General Comments:**

P2,L41: put this values in the context of equivalent changes reported in the literature, here and elsewhere.

P3,L65: You could also cite other previous works with the implementation of halogen chemistry in WRF-Chem, e.g. from Badia et al., (2019). P4,L75: Similarly, given that you compare your results with those of the CAM-Chem model, you should also cite some of the CAM-Chem studies focused on the impact of halogen chemistry over the oceans as Saiz-Lopez et al., (2014, 2023), Iglesias-Suarez et al., (2020), Li et al., (2022).

P5,L106: What version of CMAQ did you consider? Have you used and/or compared w.r.t the previous implementation of halogen chemistry in CMAQ? (e.g. Sarwar 2015).

P7,L149: You should cite and compare your methodology and emission values for anthropogenic emissions with other works to put your regional results into context of the current literature. In P8,L180 you compare with Kim et al., 2023, but should also compare with the anthropogenic emissions from Saiz-Lopez et al., 2023.

P10,L206-211: How did you validate the overall halocarbon emission inventory implemented in your model? Based on the Ordoñez et al. (2012) inventory, global scaling of chl-a bitmaps was necessary to reproduce observations. In addition, what type of diurnal profile did you apply to the emissions?

P10,L212-218: Similarly, how did you exactly implement the SSA-dehalogenation process? Note this process is very efficient and depends on many parameters that present a large spatio-temporal variability (see Ordoñez et al. 2012 and Fernandez et al., 2014). Could you please provide more details about the implementation and the net bromine flux from sea-salt.

P10,L220-223: Halogen Chemical reactions. Please, provide at least a general introduction of which are the important reference works considered in this study.

P23,L380-385: In relation to the four highlighted reactions, I can think of many other processes that could be important to evaluate: for example: i) are Cl2 or any other species assumed to be uptake into the aerosol phase and provide Cl- (aq)? Ii) Did you consider any hourly variation in the emission strength of anthropogenic halogens of HCl and Cl2 (Eq. 1) that could impact your night-time results?. What about the NO2 sources that are required for the ClNO2

formation, how is their spatial and temporal variation? (this apply also to P33,L541). If all of these were found to be irrelevant, at least a couple of sentences explaining why they are not important should be given. I completely agree that further studies are necessary to investigate the main factors causing these discrepancies (which should be highlighted in the conclusions).

P26,L434-439: The way the text is written seems to indicate that this is a result from this study, while the opposite effect between continental (polluted) and oceanic (pristine) has been previously described in the literature (e.g., Li et al., 2022, Saiz-Lopez et al., 2023). Please rephrase to make it clear that your results are in agreement with those of previous studies.

P28,L462-464: this sentence makes no sense. Please rephrase. Are you sure the SSA dehalogenation process for bromine is well implemented in your model?. P33,L545: I would expect larger impacts of bromine than chlorine over the oceans, which is not the case. Could this be possible due to a small efficiency of the SSA-dehalogenation process for bromine?

P30,Eq.8 and Eq.9: Please control the F(Ox) expression for missing production channels and explain in case some terms are not considered. For the case of D(Ox) note that the Cl+O3 term should not be considered as it results in the formation of ClO, which is part of Ox (see Saiz-Lopez et al., 2014 for a complete list of all halogen-driven OddOx loss rates).

P36,L586-594: Once again, the conclusions concentrate on the importance of halogens to improve the model representation of ClNO2 observations, which are mostly related to air-quality in polluted environments. However, the final part of the paper focus on the wider implications of halogens over continental and oceanic domains, which were not validated before. Indeed, in L602-606 you summarize model results for several species but omit mentioning the inconsistent results found for OH. In case you decide to keep all the analysis in a single paper, a detailed discussion of this important issue should be included.

**Language editing comments and Typos:**

P2,L26 and elsewhere: Please refer to halogen chemistry, not chemistries.

P22,L365: It appears → it is evident.

P35,L583: replace I by I2 inside the parenthesis.

Table 2: Check for typos and consistency in R1 and R3.

**Figures, Tables and Captions**

Table 3: given the importance of reaction R6 in your results, more details should be given in the text. Note that most model implement different versions of R1.

Figure 4: it is not clear for me if this comparison exercise considers only nightime or 24-hs model output. Given that the model is shown to underestimate Cl2 observations during the day … then it is expected that if 24-hs is considered the presented results would imply a night-time over-estimation for the EXP_CL_BR_I case. Am I right? Could you please clarify in the text?

**References**

Ordoñez et al., 2012 (https://acp.copernicus.org/articles/12/1423/2012/)

Fernandez et al., 2014 (https://acp.copernicus.org/articles/14/13391/2014/)

Saiz-Lopez et al., 2014 (https://acp.copernicus.org/articles/14/13119/2014/)

Badia et al., 2019 (https://acp.copernicus.org/articles/19/3161/2019/)

Iglesias-Suarez et al., 2020 (https://www.nature.com/articles/s41558-019-0675-6)

Li et al., 2022 (https://www.nature.com/articles/s41467-022-30456-8)

Saiz-Lopez et al., 2023 (https://www.nature.com/articles/s41586-023-06119-z)

---

## Author Comment (AC1)

**Reply to comments from Referee #1**

First of all, thank you for your valuable comments and suggestions. Following your comments, we attempt to clarify and improve the manuscript by eliminating, modifying, and adding several parts from/into the original text. The added or modified parts are painted in a blue color in the revised manuscript.

**[General Comment]**

In Kim et al., 2025, the authors add and update the halogen (Chlorine, Bromine, and Iodine) chemistry scheme in the WRF-CMAQv5.2.1 setup for the Korean peninsula region. To implement the model, they first generate new emission datasets for anthropogenic Cl, Br species and natural Iodine species for the region. They run six simulations, CTRL (no halogens), $EXP_{Cl}$(where chlorine reactions are updated), $EXP_{Cl-Br}$ (where Chlorine and Bromine are updated), $EXP_{Cl-Br-I}$ (where all three species are updated), $EXP_{CAM}$ (where Saiz-Lopez, 2014 CAM-Chem halogen scheme is used), and $EXP_{CMAQ}$ (CMAQ model with halogen chemistry by Sarwar et al., 2015). First they compare $EXP_{Cl-Br-I}$, $EXP_{CAM}$ and $EXP_{CMAQ}$ against observed $ClNO_2$, $Cl_2$ values from the KORUS-AQ campaign at two sites, Olympic Park and Mt. Taehwa. Results show that the simulation with halogen chemistry performs better in matching observed $ClNO_2$ and $Cl_2$ values, although there is uncertainty for $Cl_2$ both in terms of the simulated values and observations. Ozone changes in response to the new halogen reactions shows compensating changes with increased formation of ozone over land and increased destruction over the oceans. Finally, the impact of the halogen scheme on OH (5% increase), $HO_2$ (5.3% decrease), VOCs (5.9% decrease), NOx (2.9% increase), and HCHO (1.6% increase) is discussed. I think the paper is well-written. The structure of the paper is straight-forward and easy to understand. This study highlights the importance of having a more complete halogen chemistry in chemistry models, as it affects OH, VOCs, nitrate, sulfates, PM2.5 and $O_3$ and there is a need to understand these changes on a regional scale. Other than some changes that clarify and improve the readability of the paper, I think this paper is suitable for publication.

**[Major Suggestions]**

**Comment 1:** Can you provide more details on the model runs and how the statistics were calculated - How long was the model run for and what is the time-period of the runs? Which period were the observations taken, (summer or winter or which months) And how typical were these

observations compared to the other sites? Maybe I missed it, but can you provide a reference(s) for the campaign?

**Reply:** We ran the model from May 1 to June 12, 2016 with a 5-day spin-up days, from April 26 to April 30. These spin-up days are necessary to reduce the uncertainty of initial conditions (please refer to lines 102 and 129-130).

Statistical metrics were calculated using hourly modeled and observed concentrations with the formulas shown below:

$$RMSE = \sqrt{\frac{\sum_1^n (M-O)^2}{n}}$$

$$MB = \frac{1}{n}\sum_1^n (M-O)$$

$$IOA = 1 - \frac{\sum_1^n (M-O)^2}{\sum_1^n (|M-\bar{O}| + |O-\bar{O}|)^2}$$

M and O represent modeled and observed outputs, respectively.

Observation periods were varied by site: In the Olympic Park station, the data were produced from May 17 to June 12, while in the Taehwa Mountain, the observations were made from May 5 to June 12 (see top panel on the Figure 2). Although $ClNO_2$ was measured only at these two sites, the observed levels are comparable to those reported in the previous studies (Mielke et al., 2011; Wang et al., 2017).

We have also added two more references for the KORUS-AQ campaign (Crawford et al., 2021; Jeong et al., 2023) (please refer to line 113)

**Comment 2:** In section 3.1.1, there is an emphasis to state that adding the three-species halogen chemistry helped in bringing the model results closer to observed (lines 340-342 for example), compared to $EXP_{CMAQ}$. But most of these changes (75% or so) are solely due to updating the chlorine scheme. Have you compared $EXP_{CMAQ}$ against $EXP_{Cl}$ and $EXP_{Cl-Br}$? I think the emphasis in this section should be changed to how updating the Cl mechanism by adding new reactions and changing the parameterization of $N_2O_5$, brings down nighttime ClNO2 levels, followed by the impact of the addition of the HOBR reaction.

**Reply:** We agree with your comment, and have conducted an additional comparison among $EXP_{CMAQ}$, $EXP_{Cl}$, and $EXP_{Cl\_Br}$, as shown in the Figure R1.

[Figure]

**Figure R1.** Diurnal variations in the mixing ratios of ClNO$_2$ (unit: ppt) at (a) Olympic Park and (b) Mt.Taehwa stations during the period of the KORUS-AQ campaign. Observed values are represented by open circles (error bars indicate the standard deviation). Colored lines with shaded areas show the hourly-averaged mixing ratios of ClNO$_2$ and the corresponding standard deviation from each simulation. The black shaded area indicates the variations in the photolysis rate of ClNO2 derived from the EXP$_{Cl\_Br\_I}$ simulation

In the Fig. R1, most of the changes in simulated ClNO$_2$ levels can be attributed to the updated chlorine chemistry (EXP$_{Cl}$). We have revised Section 3.1.1 to emphasize this point, more focusing on the impacts of the updated Cl chemistry, followed by the Br-related reactions (please, refer to lines 404-407).

**Comment 3**: In section 3.1.2, can you elaborate on the sensitivity test and how it was conducted? Were all reactions considered and these four reactions stand out or was there a reason to pick only these 4? Given the uncertainties in the partitioning of N$_2$O$_5$ onto chloride containing particles in the model being a big source for the change, have you tried testing any other parameterizations other than the one used?

**Reply:** To identify the key reactions, we conducted multiple sensitivity tests with variable mixing ratios of atmospheric halogen species and reaction rates. We then excluded the reactions the contributions of which to ClNO$_2$ productions were less than ~1%.

Given the uncertainties in N$_2$O$_5$ partitioning onto chloride-containing particles, we also tested three different parameterizations. Among them, the schemes selected in this study showed the best performances in reproducing observed ClNO$_2$ levels (refer to Table R1).

**Table R1.** Statistical analysis of $ClNO_2$ mixing ratios using different $N_2O_5$ parameterizations from Bertram and Thornton (2009), Davis et al. (2008), and this study.

**(a) Olympic Park**

|  | Bertram & Thornton (2009) | Davis et al. (2008) | This study |
|---|---|---|---|
| Mean Bias (ppt) | 44.09 | 43.20 | 31.62 |
| Root Mean Square (ppt) | 200.98 | 196.24 | 179.4 |
| Index of agreement | 0.63 | 0.64 | 0.66 |
| Simulated mean (ppt) | 130.08 | 129.18 | 117.6 |

**(b) Mt.Taehwa**

|  | Bertram & Thornton (2009) | Davis et al. (2008) | This study |
|---|---|---|---|
| Mean Bias (ppt) | -22.38 | -23.57 | -31.4 |
| Root Mean Square (ppt) | 280.47 | 275.86 | 272.4 |
| Index of agreement | 0.57 | 0.57 | 0.58 |
| Simulated mean (ppt) | 136.98 | 135.79 | 128.0 |

**Comment 4**: Can you write more about the differences between $EXP_{CAM}$ and either $EXP_{CMAQ}$ or $EXP_{Cl-Br-I}$? The $ClNO_2$ is near-zero in $EXP_{CAM}$. Is this because of missing reactions in the 2014 version of CAM-Chem that was adopted and tested against here? There may have been updates and modifications to that scheme as well, so maybe a more recent version (if changed) might be useful for discussion.

**Reply:** As noted in the revised manuscript, $EXP_{CAM}$ was originally designed to simulate coastal conditions, which can limit the capability to reproduce the levels of $ClNO_2$ over NOx-rich continental regions such as the Korean Peninsula. It is likely that this may explain the near-zero $ClNO_2$ concentrations simulated from $EXP_{CAM}$. We provided more detailed explanations in the revised manuscript regarding this point (please, refer to lines 341-347 in the revised manuscript).

**Comment 5**: I wonder if there can be some discussion on uncertainties in the simulations (could be in the supplementary), especially considering that some of the changes seem to be on the smaller side - such as changes in ozone (in terms of percentage changes). I appreciate that is because of the competing reactions that affect the formation and destruction of ozone. But are these changes significant?

**Reply:** We agree with your point. Incorporating the full halogen chemistries and processes into the CMAQ simulations may be very challenging due to several following uncertainties: (i) high spatio-temporal variability in halogen emissions, (ii) omissions of potential reactions, and (iii) limited availability and accuracy of observational data. We added these limitations in our revised manuscript (please, refer to lines 689-695).

Although the changes in $O_3$ appear to be small in South Korea due to the competing effects between halogen-induced production and destruction, it is still important to incorporate sophisticated halogen reactions in modeling simulations. Halogen chemistries not only affects ozone levels but also radical chemistry in the atmosphere, potentially enhancing the lifetime of greenhouse gases such as $CH_4$ (Li et al., 2022; A.Saiz-Lopez et al., 2023). This is the necessity to highlight the detailed halogen reactions in the atmospheric models.

**Comment 6**: Is the updated version of the model code and the results going to be available to the public? If so, can you give the location for that as well in addition to the CMAQ web page.

**Reply:** This study was conducted as a part of a national project to develop the Korean Air chemistry Modeling System (K_ACheMS). Model code is not publicly available at this moment, but it can be provided upon request.

**[Minor Suggestions]**
**Comments 1:** Line 83: Change examines to examine.
**Reply:** We revised it.

**Comments 2:** Line 274: CCHO should be CH3CHO.
**Reply:** We revised it.

**Comments 3:** Line 381: Maybe use "Future" instead of "Further".
**Reply:** Thank you for your correction. We changed it.

**Comments 4:** Line 578: I think "attempted to incorporate" can be modified because CMAQ modeling systems already exist with some halogen processes. Maybe being clearer that you added and updated three halogen species reactions in the CMAQ modeling system would be better.
**Reply:** We clarified this point in the revised manuscript. Please check out lines 230-233.

**Comments 5:** Figure 3b: Perhaps this is a rounding error, but the sum of the percentages in the pie chart add up to 100.1%.
**Reply:** We corrected this (please, see Fig. 3b).

**Comments 6:** Figure 3: In the caption, can you add that the grey regions are night-time?
**Reply:** We added it (please, see Fig. 3).

[Figure]

**Figure 3.** Contributions of halogen reactions to the mixing ratios of $ClNO_2$ in the $EXP_{CMAQ}$ and $EXP_{Cl\_Br\_I}$ simulations at (a and b) Olympic Park and (c and d) Mt. Taehwa stations during the period of KORUS-AQ campaign. Stacked bars and pie charts show the contributions from four halogen reactions to the mixing ratios of $ClNO_2$. Grey-shaded areas represent nighttime (18:00-06:00 local standard time).

**Comments 7** Figure 5: What do the DIFF's stand for? Why is it MAX for the top two and MIN for the bottom panel? Are the standard deviations and error bars for the observed?
**Reply:** The 'DIFF' values represent the difference in average concentrations between simulations during the analysis period. The 'MIN' and 'MAX' mean the minimum and maximum of these average differences. We though that these are a bit out of context. Thus, we removed it.

To further clarify the meaning of DIFF and to avoid any confusion regarding the standard deviation and error bars, we have updated the Figure caption (please, refer to Fig. 6).

[Figure]

**Figure 6.** Diurnal variations in the mixing ratios of $O_3$ from CTRL (black circles) and $EXP_{Cl\_Br\_I}$ (red circles) simulations, together with observed mixing ratios of $O_3$ (OBS; white circles) at (a) Olympic Park, (b) Mt. Taehwa, and (c) Bangnyung stations during the period of the KORUS-AQ campaign. Error bars and shaded areas indicate the standard deviations of observed and modeled $O_3$, while the grey-shaded areas show the nighttime. $DIFF_{mean}$ represents the difference in the averaged mixing ratios of $O_3$ between $EXP_{Cl\_Br\_I}$ and CTRL simulations.

**Comments 8:** Figure 8: Can you add the mean values of changes within or in top of the panel each of the species & simulations? It would be nice to see what the changes are for each of these experiments within the figure itself. At least for the full halogen process panel.
**Reply:** We added them in revised Fig. 9.

[Figure]

**Figure 9.** Summaries of the impacts of chlorine processes ($EXP_{Cl}$ − CTRL), bromine processes ($EXP_{Cl\_Br}$ − $EXP_{Cl}$), iodine processes ($EXP_{Cl\_Br\_I}$ − $EXP_{Cl\_Br}$), and full halogen processes ($EXP_{Cl\_Br\_I}$ − CTRL) on the mixing ratios of OH, $HO_2$, HCHO, VOCs, and $NO_x$, respectively, during the period of KORUS-AQ campaign. Also, $Land\_DIFF_{mean}$ and $Ocean\_DIFF_{mean}$ indicate the differences in the averaged mixing ratios of each species between $EXP_{Cl\_Br\_I}$ and CTRL simulations, over land and ocean, respectively.

**Reference cited in this response:**

Mielke, L. H., Furgeson, A., & Osthoff, H. D. (2011). Observation of ClNO2 in a mid-continental urban environment. *Environmental Science & Technology*, *45*(20), 8889-8896.

Wang, X., Wang, H., Xue, L., Wang, T., Wang, L., Gu, R., ... & Wang, W. (2017). Observations of N2O5 and ClNO2 at a polluted urban surface site in North China: High N2O5 uptake coefficients and low ClNO2 product yields. *Atmospheric environment*, *156*, 125-134.

Crawford, J. H., Ahn, J. Y., Al-Saadi, J., Chang, L., Emmons, L. K., Kim, J., ... & Kim, Y. P. (2021). *The Korea–United States Air Quality (KORUS-AQ) field study, Elem. Sci. Anth., 9, 00163*.

Jeong, D., Seco, R., Gu, D., Lee, Y., Nault, B. A., Knote, C. J., Mcgee, T., Sullivan, J. T., Jimenez, J. L., Campuzano-Jost, P., Blake, D. R., Sanchez, D., Guenther, A. B., Tanner, D.,

Huey, L. G., Long, R., Anderson, B. E., Hall, S. R., Ullmann, K., Shin, H., Herndon, S. C., Lee, Y., Kim, D., Ahn, J., and Kim, S.: Integration of airborne and ground observations of nitryl chloride in the Seoul metropolitan area and the implications on regional oxidation capacity during KORUS-AQ 2016, Atmos. Chem. Phys., 19, 12779–12795, https://doi.org/10.5194/acp-19-12779-2019, 2019.

Li, Q., Fernandez, R.P., Hossaini, R. *et al.* Reactive halogens increase the global methane lifetime and radiative forcing in the 21[st] century. *Nat Commun* **13**, 2768 (2022). https://doi.org/10.1038/s41467-022-30456-8

Saiz-Lopez, A., Fernandez, R.P., Li, Q. *et al.* Natural short-lived halogens exert an indirect cooling effect on climate. *Nature* **618**, 967–973 (2023). https://doi.org/10.1038/s41586-023-06119-z

---

## Author Comment (AC2)

**Reply to comments from Referee #2**

First of all, thank you for your valuable comments and suggestions. Following your comments, we attempt to clarify and improve the manuscript by eliminating, modifying, and adding several parts from/into the original text. The added or modified parts are painted in a blue color in the revised manuscript.

**[General Comment]**

The paper presents the implementation of halogen chemistry in the CMAQ model, including a comprehensive representation of chlorine, bromine and iodine sources and chemistry. Once the model updates are described, the study focus on reproducing ClNO2 and O3 observations over the Korean Peninsula during the KORUS-AQ campaign. The results focus on the improvements in the model-observation comparisons, including the index of agreement and other statistical parameters, particularly over two inland locations within a polluted / semi-polluted environment. Then, they identify the four main reactions in the model accounting for the largest fraction of the improvement. Based on this, the authors evaluate the influence of air quality in the whole modeled domain, highlighting the major and sometimes opposite differences observed over continental and oceanic domains, and provide general conclusions about the benefits of considering halogen chemistry in the study.

First of all, I would like to recognize the efforts from the group to implement their own version of halogen chemistry in CMAQ, which is of major importance as the community needs more modeling studies focused on the halogen influence on atmospheric chemistry and climate. However, I believe the current version of the work does not allow reaching a firm conclusion of the results obtained, probably because the paper attempts to address all at once the complete technical implementation, the observational improvements achieved over continental locations, and the overall impact and influences over different regions. While I leave for the authors to decide if it is convenient to present all of these developments within a single work or to partially split into independent companion papers, I recommend the authors to address the following major comments and submit a revised version for further consideration.

**Reply:** This work may try to address too many topics. However, we decided to comprehensively cover all these contents within a single paper. We have revised the manuscript based on two reviewer's comments, making this paper clearer.

**[Major Comments]**

**Comment 1:** P4, L93: In the introduction you clearly state that "We then investigate the formation of ozone using the new halogen processes", which is what the paper focus first … but then at the end the work attempts to provide much wider conclusions of the halogen influence on atmospheric

chemistry and air quality. You may want to focus here on ozone production driven by ClNO2 chemistry, and leave the more general discussion for a companion paper.

**Reply:** The main objective of this study is to implement updated halogen chemistry and halogen processes in the framework of CMAQ model, and then assess their implications in the atmospheric chemistry. This objective is described in the lines 91-107 in the revised manuscript. Chemistry related to nitryl chloride ($ClNO_2$) is just a part of the entire work. We mentioned $ClNO_2$ many times in the manuscript, because it is only the halogen species measured during the KORUS-AQ campaign. Thus, without $ClNO_2$, it is almost impossible to confirm that our model has been correctly developed. To avoid the impression that our model evaluation is restricted to ClNO2, we added a new Section 3.1.4. This section includes indirect comparisons of modeled BrO and IO levels with values reported in the previous studies.

**Comment 2:** P5, L111: Comparison with ClNO2 observations are of mayor importance for this study. Indeed, it might be worth to mention in the title. Current title and abstract give the impression of a general halogen chemistry development, while the work mostly focus on ClNO2 and its role for ozone production.

**Reply:** As mentioned in our response to your Comment 1, we have revised the manuscript to diminish the impression that current study solely focuses on $ClNO_2$. We added new Section 3.1.4. to provide a broader evaluation of the halogen chemistry by comparing simulated the mixing ratios of BrO and IO with those reported in the previous studies (please refer to lines 24-37, 99-107, 456-480, and 670-675).

[Figure]

**Figure 5.** Comparison between modeled and observed mixing ratios of atmospheric (a) BrO and (b) IO. Both the observed and modeled mixing ratios were obtained from the previous studies.

**Comment 3**: P20,L326-328: Your results for EXP_CAM are not surprising to me as the heterogeneous ClNO2 formation through N2O5 was not considered in Saiz-Lopez et al., (2014), which is your reference. However, it should be mentioned that those studies focused on oceanic and pristine conditions, and not polluted areas with high NOx and inland chlorine emissions. Indeed, further research from the group lead by Dr. Saiz-Lopez considered enhanced HCl and ClNO2 production within continental areas (see for example Li et al. 2022), which clearly showed important implications when anthropogenic chlorine sources are considered. Therefore I recommend including the complete chlorine scheme from CAM-Chem in your analysis to avoid reaching erroneous conclusion, or at least to clarify why your EXP_CAM simulations do not reproduce ClNO2 observations.

**Reply:** To avoid possible confusion of readers, we added a more detailed explanation into the revised manuscript regarding the limitations of the $EXP_{CAM}$ simulation in capturing the levels of $ClNO_2$ under high NOx and inland conditions. Please, check out the lines 341-347 in the revised manuscript.

**Comment 4**: P22,L356-367: Of all four dominant process mentioned here, R20 in Table 1 (ClO + ClO) is particularly surprising to me, as this reaction is typically considered in stratospheric ozone depletion, but not for boundary layer studies. How do you explain that the surface observations are sensible to this reaction? Is it because of the subsequent Cl2 photolysis? Is due to the coupling of ClO + NO2? Can you explain how this reaction can contribute to ClNO2 formation during the night, when ClO abundance is zero and in addition, any Cl2 formed would not be photolyzed until the next morning?

**Reply:** In our model, we set the rate constant for R20 at a value 10 times lower than that used in the original CMAQ model (refer to Table R2 shown below). This results in slower ClO removal and slightly higher ClO concentrations in the boundary layer.

**Table R2.** Comparison of reaction rate presented in Table 1 between $EXP_{CMAQ}$ and this study

| | Reaction | $EXP_{CMAQ}$ | This study ($EXP_{Cl\_Br\_I}$) |
|---|---|---|---|
| R20 in Table 1 | $ClO + ClO \xrightarrow{k_1} 0.29Cl_2 + 1.42Cl$ | $ClO + ClO \xrightarrow{k_1} 0.29Cl_2 + 1.42Cl$
 $k_1 = 1.25\times10^{-11} e^{-1960/T}$ | $ClO + ClO \xrightarrow{k_1} Cl_2$
 $k_1 = 1.0\times10^{-12} e^{-1590/T}$ |

Although ClO concentrations are generally low at night, the small increase may enhance the formation of $ClONO_2$ via the $ClO + NO_2$ reaction. The formation of such reservoir species ($ClONO_2$) could indirectly limit $ClNO_2$ production by reducing the availability of reactive nitrogen. We have revised the manuscript to clarify this mechanism (please refer to lines 393-400).

**Comment 5**: P33,L547-562: I follow the explanation about the changes in the OH/HO2 partitioning, but oceanic SLH (particularly iodine) has been clearly shown to reduce the OH abundance (not increase it), and consequently to increase the CH4 burden and lifetime (Li et al., 2022). Your results over the ocean seems to contradict that. How could this be? Indeed, the null cycle mentioned in your work results in a shift of partitioning from OH to HO2 (which is fine), but the total OH abundance is controlled by O3 + hv --> O1D. Given that oceanic halogens reduce O3, there is less O1D and therefore OH formation should decrease. Have you discarded any influence from the BC affecting the overall results? Note that O3 changes for the EXP_Cl_Br_I - EXP_ctrl are in line with Li et al. (2022) for oceanic domains, but the OH changes are not consistent with the changes in O3 (unless I missed something).

**Reply:** Previous global-scale studies have shown that oceanic iodine chemistry generally reduces OH, mainly because it suppresses $O_3$ production and thus $O(^1D)$ production (e.g., Li et al., 2022). However, during the short-term episodes such as the KORUS-AQ campaign, our results indicate that OH mixing ratios can increase despite the overall reduction in $O_3$. One possible explanation about this is that halogen-mediated reactions, particularly the reaction of IO with $HO_2$, and the photolysis of HOI, may efficiently regenerate OH radicals. These processes can partially offset, the OH loss associated with lower $O_3$ and $O(^1D)$ production. Similar behavior has also been reported in the previous studies (e.g., Saiz-Lope et al., 2012; Stone et al., 2018; Mahajan et al., 2021). To clarify this, we added a discussion into the revised manuscript, emphasizing that our results are based on limited-time campaign (please, refer to lines 610-619).
Also, the influence of black carbon (BC) was not taken into account in this study. Future sensitivity tests may be necessary to more completely assess its potential impacts on halogen-radical interactions.

**[General Comments]**
**Comments 1:** P2,L41: put this values in the context of equivalent changes reported in the literature, here and elsewhere.
**Reply:** To address this point, we revised Section 3.4 including comparisons analysis with relevant previous studies (please, refer to lines 625-629 and 632-634).

**Comments 2:** P3,L65: You could also cite other previous works with the implementation of halogen chemistry in WRF-Chem, e.g. from Badia et al., (2019). P4,L75: Similarly, given that you compare your results with those of the CAM-Chem model, you should also cite some of the CAM-Chem studies focused on the impact of halogen chemistry over the oceans as Saiz-Lopez et al., (2014, 2023), Iglesias-Suarez et al., (2020), Li et al., (2022).
**Reply:** We mentioned (cited) those references in our revise manuscript (See line 75).

**Comments 3:** P5,L106: What version of CMAQ did you consider? Have you used and/or compared w.r.t the previous implementation of halogen chemistry in CMAQ? (e.g. Sarwar 2015).

**Reply:** In this study, we used the US EPA CMAQ version 5.2.1. In addition to the newly developed $EXP_{Cl\_Br\_I}$ model, we implemented the halogen chemistry scheme developed by Sarwar et al. (2015) again in the framework of CMAQ v5.2.1 ($EXP_{CMAQ}$). This was used as a reference for model evaluation, particularly in the comparison of $ClNO_2$ mixing ratios (please, refer to lines 230-233 and 314-319).

**Comments 4:** P7,L149: You should cite and compare your methodology and emission values for anthropogenic emissions with other works to put your regional results into context of the current literature. In P8,L180 you compare with Kim et al., 2023, but should also compare with the anthropogenic emissions from Saiz-Lopez et al., 2023.

**Reply:** Previous studies of Saiz-Lopez et al., 2023; Fu et al. 2020 provided valuable insights into global- and China-scale anthropogenic halogen emissions. However, we think that comparison of the previous emission estimates with our one may have limited relevance due to large differences in spatial scale and emission inventories.
Therefore, we included additional comparisons with other studies conducted over the same regions and time scales as our study (please, refer to lines 185-188).

**Comments 5:** P10,L206-211: How did you validate the overall halocarbon emission inventory implemented in your model? Based on the Ordoñez et al. (2012) inventory, global scaling of chl-a bitmaps was necessary to reproduce observations. In addition, what type of diurnal profile did you apply to the emissions?

**Reply:** To minimize uncertainty in chl-a bitmaps over East Asia, we utilized satellite-derived highly-resolved chl-a data from the GOCI instrument. The geostationary GOCI sensor was specially designed for observing ocean colors in the East Asia (Park et al. 2015). Based on these data, spatial distributions and relative proportions of $Br_y$ and $I_y$ were calculated. Some results are presented in Figure R1 shown below. The estimated values fall within the ranges reported in the previous studies (e.g., Li et al., 2020; Huang et al., 2021, Sherwen et al. 2017).

[Figure]

**Figure R1.** Spatial mixing ratios of the (a) $Br_y$ and (d) $I_y$ over the ocean and the pie chart of $Br_y$ (b and c) and $I_y$ (e and f) during daytime and nighttime, respectively. The numbers represent the ratios of each halogen species during the period of the KORUS-AQ campaign.

Regarding the temporal variation, default diurnal profile provided by the CMAQ model was used (refer to the diurnal scaling factors $f_{dp}$ in Eq.2). The hourly scaling factors over 24 hours are: 0.032, 0.032, 0.032, 0.033, 0.034, 0.036, 0.039, 0.044, 0.051, 0.057, 0.062, 0.064, 0.062, 0.057, 0.051, 0.051, 0.044, 0.039, 0.034, 0.033, 0.032, 0.032, 0.032, 0.032. These diurnal scale factors were uniformly applied to all the halogen species.

**Comments 6:** P10,L212-218: Similarly, how did you exactly implement the SSA-dehalogenation process? Note this process is very efficient and depends on many parameters that present a large spatio-temporal variability (see Ordoñez et al. 2012 and Fernandez et al., 2014). Could you please provide more details about the implementation and the net bromine flux from sea-salt.

**Reply:** Good point! The SSA dehalogenation is a key and highly uncertain process in the halogen chemistry, having strong dependence on various spatio-temporal parameters. To account for this process, we adopted an approach used by Sarwar et al. (2015) as shown in below.

- $\mathbf{E_{Br_2}}$: $0.864 \times (O_F + S_F) \times A_{GC} \times 0.965 \times 10^{-16} \times U_{10} \times (0.38 + 0.054 \times SSTC) \times \rho_{SSA} \times R_a \times DF/MW_{Br_2}$

- $\mathbf{E_{I_2}}$: $(O_F + S_F) \times A_{GC} \times 1.16 \times 10^{-14} \times [O_3] \times [I^-_{(aq)}]^{1.3} \times (1.74 \times 10^9 - (6.54 \times 10^8 \times \ln ws))$

- $\mathbf{E_{HOI}}$: $(O_F + S_F) \times A_{GC} \times 1.16 \times 10^{-14} \times [O_3] \times (4.15 \times 10^5 \times ([I^-_{(aq)}]^{0.5}/ws) - (20.6/ws) - 23600 \times [I^-_{(aq)}]^{0.5})$

*$O_F$: ocean zone fraction, $S_F$: surf zone fraction, $A_{GC}$: grid cell area, SSTC: sea surface temperature, $U_{10}$: 10m wind speed, $\rho_{SSA}$: dry SSA density, $R_a$: sea-salt Br/NaCl ratio, DF: bromine depletion factor, $MW_{Br_2}$: molecular weight of $Br_2$, ws: wind speed*

We added the description of the SSA-dehalogenation process into the revised manuscript (please, check out lines 220-228).
The net bromine flux from SSA is shown in Figure R1 (again, refer to the response to Comment 5). As described in Fig. R1, the flux levels are within the range reported in the previous studies (Li et al., 2020; Huang et al., 2021, Sherwen et al. 2017).

**Comments 7:** P10,L220-223: Halogen Chemical reactions. Please, provide at least a general introduction of which are the important reference works considered in this study.
**Reply:** We tried to provide a clearer explanation regarding this (please, refer to lines 230-233).

**Comments 8:** P23,L380-385: In relation to the four highlighted reactions, I can think of many other processes that could be important to evaluate: for example: i) are Cl2 or any other species assumed to be uptake into the aerosol phase and provide Cl- (aq)? Ii) Did you consider any hourly variation in the emission strength of anthropogenic halogens of HCl and Cl2 (Eq. 1) that could impact your night-time results?. What about the NO2 sources that are required for the ClNO2formation, how is their spatial and temporal variation? (this apply also to P33,L541). If all of these were found to be irrelevant, at least a couple of sentences explaining why they are not important should be given. I completely agree that further studies are necessary to investigate the main factors causing these discrepancies (which should be highlighted in the conclusions).
**Reply:** In this study, we carried out a series of sensitivity tests to evaluate the influences of key halogen-related reactions by comparing simulations with and without these processes. However, as the reviewer correctly pointed out, several other factors may also contribute to the discrepancies we found.
First, this study did not apply diurnal variation to anthropogenic emissions of HCl and $Cl_2$, which may affect nighttime $ClNO_2$ formation. Second, the model tends to underestimate nighttime $NO_2$ compared to observations (see Figure R2), which can potentially limit $ClNO_2$ production. Also, particulate uptake of $Cl_2$ and detailed NO2 variations were not explicitly considered. Regarding these points, we added their potential importance and the need for further investigation in the revised manuscript (please refer to lines 411-415).

[Figure]

**Figure R2.** Diurnal variations of the mixing ratios of $NO_2$ from the observation (blue line) and the CMAQ model (red line) during the period of the KORUS-AQ campaign.

**Comments 9:** P26,L434-439: The way the text is written seems to indicate that this is a result from this study, while the opposite effect between continental (polluted) and oceanic (pristine) has been previously described in the literature (e.g., Li et al., 2022, Saiz-Lopez et al., 2023). Please rephrase to make it clear that your results are in agreement with those of previous studies.

**Reply:** We tried to revise the manuscript to emphasize the consistency between our results and those from the previous studies (please refer to lines 536-539).

**Comments 10:** P28,L462-464: this sentence makes no sense. Please rephrase. Are you sure the SSA dehalogenation process for bromine is well implemented in your model?. P33,L545: I would expect larger impacts of bromine than chlorine over the oceans, which is not the case. Could this be possible due to a small efficiency of the SSA-dehalogenation process for bromine?

**Reply:** Around the Korean peninsula, the interactions between VOCs and $NO_x$ are complex. To better represent bromine chemistry, we should incorporate reactions between bromine radicals and VOCs (refer to R46-R56 in Table 4), which were not included in the original CMAQ model. In addition, anthropogenic emissions of HBr and $Br_2$ were newly added to support these reactions. Without considering these VOC-related reactions, bromine radicals would primarily lead to ozone depletion (e.g., Br + $O_3$), but the inclusion of fast reactions such as Br + VOC can contribute to ozone production, partially compensating for the ozone loss. We have revised the manuscript to clarify this mechanism and its implications (please, refer to lines 520-523).

**Comments 11:** P30,Eq.8 and Eq.9: Please control the F(Ox) expression for missing production channels and explain in case some terms are not considered. For the case of D(Ox) note that the Cl+O3 term should not be considered as it results in the formation of ClO, which is part of Ox (see Saiz- Lopez et al., 2014 for a complete list of all halogen-driven OddOx loss rates).
**Reply:** We revised the manuscript, updating the description of the $O_x$ family (please, refer to lines 552-556).

$$D(O_x) = k_{NO_2+OH}[NO_2][OH] + k_{O_3+VOC}[O_3][VOC] + k_{O(1D)+H_2O}[O(^1D)][H_2O]$$
$$+ k_{O_3+OH}[O_3][OH] + k_{O_3+HO_2}[O_3][HO_2] + k_{RO_2+NO_2}[RO_2][NO_2]$$
$$+ (k_{XO+O_3}[O_3] + k_{XO+HO_2}[HO_2] + k_{XO+O(3p)}[O(^3P)])[XO]$$
$$+ k_{ClO+NO_2}[NO_2][ClO] + k_{XO+XO}[XO]^2 + k_{NO_3+VOC}[NO_3][VOC]$$
$$+ 3k_{het}[N_2O_5] \hspace{4cm} \text{(Eq. 9)}$$

**Comments 12:** P36,L586-594: Once again, the conclusions concentrate on the importance of halogens to improve the model representation of ClNO2 observations, which are mostly related to air-quality in polluted environments. However, the final part of the paper focus on the wider implications of halogens over continental and oceanic domains, which were not validated before. Indeed, in L602-606 you summarize model results for several species but omit mentioning the inconsistent results found for OH. In case you decide to keep all the analysis in a single paper, a detailed discussion of this important issue should be included.
**Reply:** We tried to revise our manuscript, restricting the contents with a more detailed discussion and with more emphasis on the key issues (please, refer to lines 29-34; 91-107; 456-480; 610-619; 670-675; and 689-695).

**[Language editing comments and Typos]**
**Comments 1:** P2,L26 and elsewhere: Please refer to halogen chemistry, not chemistries.
**Reply:** We changed them.

**Comments 2:** P22,L365: It appears, it is evident.
**Reply:** We revised it.

**Comments 3:** P35,L583: replace I by I2 inside the parenthesis.
**Reply:** Thank you! We corrected this.

**Comments 4:** Table 2: Check for typos and consistency in R1 and R3.
**Reply:** We corrected typos.

**[Figure, Tables and Captions]**

**Comments 1:** Table 3: given the importance of reaction R6 in your results, more details should be given in the text. Note that most model implement different versions of R1.

**Reply:** We added a more detailed explanation about reaction R6 in the revised manuscript to highlight its role and importance in the atmospheric halogen chemistry (please refer to lines 402-404).

Regarding the reaction R1, we conducted a separate sensitivity analysis, comparing different parameterizations. The results, shown in Table R1, indicate that the parameterization adopted in this study provided the best agreement with observations.

**Table R1.** Statistical analysis of $ClNO_2$ mixing ratios using different $N_2O_5$ parameterizations from Bertram and Thornton (2009), Davis et al. (2008), and this study.

**(a) Olympic Park**

|  | Bertram & Thornton (2009) | Davis et al. (2008) | This study |
|---|---|---|---|
| Mean Bias (ppt) | 44.09 | 43.20 | 31.62 |
| Root Mean Square (ppt) | 200.98 | 196.24 | 179.4 |
| Index of agreement | 0.63 | 0.64 | 0.66 |
| Simulated mean (ppt) | 130.08 | 129.18 | 117.6 |

**(b) Mt.Taehwa**

|  | Bertram & Thornton (2009) | Davis et al. (2008) | This study |
|---|---|---|---|
| Mean Bias (ppt) | -22.38 | -23.57 | -31.4 |
| Root Mean Square (ppt) | 280.47 | 275.86 | 272.4 |
| Index of agreement | 0.57 | 0.57 | 0.58 |
| Simulated mean (ppt) | 136.98 | 135.79 | 128.0 |

**Comments 2:** Figure 4: it is not clear for me if this comparison exercise considers only nighttime or 24-hs model output. Given that the model is shown to underestimate Cl2 observations during the day ... then it is expected that if 24-hs is considered the presented results would imply a nighttime over-estimation for the EXP_CL_BR_I case. Am I right? Could you please clarify in the text?

**Reply:** The analysis, presented in Fig. 4, is based on 24-hour model outputs. We revised the manuscript to clearly describe and discuss this diurnal behavior and its implications for the interpretation of the results (please, refer to lines 439-441).

**Reference cited in this response:**

Li, Q., Fernandez, R.P., Hossaini, R. *et al.* Reactive halogens increase the global methane lifetime and radiative forcing in the 21$^{st}$ century. *Nat Commun* **13**, 2768 (2022). https://doi.org/10.1038/s41467-022-30456-8

Saiz-Lopez, A., Fernandez, R. P., Ordóñez, C., Kinnison, D. E., Gómez Martín, J. C., Lamarque, J.-F., and Tilmes, S.: Iodine chemistry in the troposphere and its effect on ozone, Atmos. Chem. Phys., 14, 13119–13143, https://doi.org/10.5194/acp-14-13119-2014, 2014.

Saiz-Lopez, A., & von Glasow, R. (2012). Reactive halogen chemistry in the troposphere. *Chemical Society Reviews*, *41*(19), 6448-6472.

Stone, D., Sherwen, T., Evans, M. J., Vaughan, S., Ingham, T., Whalley, L. K., Edwards, P. M., Read, K. A., Lee, J. D., Moller, S. J., Carpenter, L. J., Lewis, A. C., and Heard, D. E.: Impacts of bromine and iodine chemistry on tropospheric OH and HO$_2$: comparing observations with box and global model perspectives, Atmos. Chem. Phys., 18, 3541–3561, https://doi.org/10.5194/acp-18-3541-2018, 2018.

Mahajan, A. S., Li, Q., Inamdar, S., Ram, K., Badia, A., and Saiz-Lopez, A.: Modelling the impacts of iodine chemistry on the northern Indian Ocean marine boundary layer, Atmos. Chem. Phys., 21, 8437–8454, https://doi.org/10.5194/acp-21-8437-2021, 2021.

Saiz-Lopez, A., Fernandez, R.P., Li, Q. *et al.* Natural short-lived halogens exert an indirect cooling effect on climate. *Nature* **618**, 967–973 (2023). https://doi.org/10.1038/s41586-023-06119-z

Fu, X., Wang, T., Wang, S., Zhang, L., Cai, S., Xing, J., & Hao, J. (2018). Anthropogenic emissions of hydrogen chloride and fine particulate chloride in China. *Environmental science & technology*, *52*(3), 1644-1654.

Kim, H., Park, R. J., Kim, S., Jeong, J. I., Jeong, D., Fu, X., & Cho, S. (2023). Effect of nitryl chloride chemistry on air quality in South Korea during the KORUS-AQ campaign. *Atmospheric Environment*, *312*, 120045.

Jo, H. Y., Park, J., Heo, G., Lee, H. J., Jeon, W., Kim, J. M., ... & Kim, C. H. (2023). Interpretation of the effects of anthropogenic chlorine on nitrate formation over northeast Asia during KORUS-AQ 2016. *Science of The Total Environment*, *894*, 164920.

Park, M. O., Shin, W. C., Son, Y. B., & Noh, T. G. (2015). Spatial Variability of in situ and GOCI and MODIS Chlorophyll and CDOM in Summer at the East Sea. *Journal of the Korean Society of Marine Environment & Safety*, *21*(4), 327-338.

Li, Q., Badia, A., Wang, T., Sarwar, G., Fu, X., Zhang, L., ... & Saiz-Lopez, A. (2020). Potential effect of halogens on atmospheric oxidation and air quality in China. *Journal of Geophysical Research: Atmospheres*, *125*(9), e2019JD032058.

Huang, Y., Lu, X., Fung, J. C., Sarwar, G., Li, Z., Li, Q., ... & Lau, A. K. (2021). Effect of bromine and iodine chemistry on tropospheric ozone over Asia-Pacific using the CMAQ model. *Chemosphere*, *262*, 127595.

Sherwen, T., Evans, M. J., Carpenter, L. J., Schmidt, J. A., & Mickley, L. J. (2017). Halogen chemistry reduces tropospheric O 3 radiative forcing. *Atmospheric Chemistry and Physics*, *17*(2), 1557-1569.

---

## Author Response (AR2)

**Reply to comments**

I sincerely appreciate your valuable suggestions. Based on your comments, we incorporated the recommended revisions into our manuscript. The changes are highlighted in blue in the revised manuscript.

**[General Comment]**

I would like to thank the authors for addressing most of the concerns raised. I just have a couple of minor issues/suggestions that can be incorporated.

**Comment 1:** Regarding Reviewer1 Comment2: In section 3.1.1, there is an …", the authors point to the EXP_cl and EXP_cl_br lines in Fig. R1, but I do not see those lines in the figure.

**Reply:** We apologize for the confusion caused by the inconsistent labelling. In Fig. R1, $EXP_{Cl}$ and $EXP_{Cl\_Br}$ are shown as green and yellow lines, respectively, together with the results from CTRL, $EXP_{CMAQ}$, and observations. We revised the Fig. R1 to ensure that all labels are accurate and clearly correspond to the experimental cases.

[Figure]

**Figure R1.** Diurnal variations in the mixing ratios of $ClNO_2$ (unit: ppt) at (a) Olympic Park and (b) Mt.Taehwa stations during the period of the KORUS-AQ campaign. Observed values are represented by open circles (error bars indicate the standard deviation). Colored lines with shaded areas show the hourly-averaged mixing ratios of $ClNO_2$ and the corresponding standard deviation from each simulation. The black shaded area indicates the variations in the photolysis rate of $ClNO2$ derived from the $EXP_{Cl\_Br\_I}$ simulation

**Comment 2:** Regarding the point of using EXP_CAM from Saiz-Lopez et al., 2014 - it might be worth adding a line that newer versions of the model do exist that account for the HCl and ClNO2 production - with the reference. That way, the reader is fully aware that that model has also been updated since.

**Reply:** We added a sentence to clarify that more recent version of the model (please, refer to lines 347-348), along with the relevant reference.

**Comment 3:** The authors could consider adding a line or two on the ranges of mixing ratios and reactions rates that were perturbed for the sensitivity tests.

**Reply:** Thank you for your suggestions. However, we believe that Table S2 provides the ranges of reaction rates used in the sensitivity test.

**Comment 4:** This is very minor, but I noticed the authors changed HOCl in Table 3 (2?) to H0Cl. If so, then there are other variables in the table that need to be changed as well such as HOBr (r8) and HOI (r15). And in Table 4 (3?).

**Reply:** To clarify, it is noted that the character 'O' in HOCl in Table 2 was not incorrectly written as the number '0'; rather, it was a matter of font style. Accordingly, we have unified the font style and applied the same formatting to Tables 2 and 3.